# LNE-BASED BLOCKING GENERATION AGAINST DATA CONTAMINATION ON LARGE LANGUAGE MODELS

## ABSTRACT

Data contamination gradually becomes inevitable during the development of large language models (LLMs), meaning the training data commonly integrates those evaluation benchmarks unintentionally. This subsequently makes it hard to benchmark LLMs fairly. This paper introduces a novel approach called LNE-Blocking for contamination detection and contamination mitigation evaluation. For the first component, Length Normalized Entropy (LNE) reports a SOTA performance for contamination detection. On this basis, LNE-Blocking reports a SOTA performance for contamination mitigation evaluation by applying LNE to adjust the intensity of blocking operation, specialized to suppressing the maximum value of output candidates during the generation process. We conduct extensive experiments on both contamination detection and contamination mitigation evaluation tasks. The results indicate that LNE and LNE-Blocking achieve an obvious SOTA performance. Simultaneously, LNE-Blocking is robust across tasks and models of different contamination level, reducing computational costs by nearly 25x compared to previous methods. We hope our method will open new research avenues on data contamination for LLMs. We plan to release the resources upon publication of this work to facilitate future work.

## 1 INTRODUCTION

In the era of fierce development with large language models (LLMs), it has been a popular research topic across many areas such as chain-of-thought reasoning (Wang et al., 2023; Wei et al., 2024), machine translation (Lu et al., 2023; Zhu et al., 2024a), code generation (Li et al., 2023a; Zhang et al., 2023), and even spatial reasoning (Hu et al., 2024). Despite the fact that LLMs are usually strong on many tasks, Natural Language Processing (NLP) practitioners secretly face a common problem called data contamination when conducting NLP research and engineering which frequently relies on benchmark data that could be potentially contaminated in LLMs.

Data contamination, also called data leakage, occurs when the test data is inadvertently included in the model's training data (Magar & Schwartz, 2022; Golchin & Surdeanu, 2024). This causes the model to perform exceptionally well on the leaked test data. Due to the immense scale and diverse origins of the pre-trained datasets used for LLMs, they are more vulnerable to data contamination. This can be divided into two main scenarios: 1) Existing benchmark datasets are more prone to leakage because of extensive text quotations and code reuse present in the LLMs' training data. 2) For emerging benchmark datasets, newly created test data may already be included in the constantly expanding training data of LLMs, as the details of these training datasets are often unknown.

As a result, preventing benchmark data contamination in LLMs becomes highly challenging. This prevents NLP developers and researchers from honestly judging the LLMs. With the growth of the training set which is partially synthetic or automatically crawled from the internet, the problem of data contamination is gradually unavoidable, even when the LLM developers do not intentionally do it. Hence, our research question is particularly important: how can we accurately detect data contamination and how can we accurately assess model performance even when it is contaminated?

To remediate the problem of data contamination, previous research mainly focused on how to detect whether the data is leaked in the models rigorously (Deng et al., 2024; Shi et al., 2024; Elazar et al., 2024). One less studied yet important problem is to quantitatively assess the genuine perfor-

mance of the models, even if they are contaminated (Zhu et al., 2024b; Dong et al., 2024; Ye et al., 2024). Such research, knowned as contamination mitigation evaluation, could be quite useful as LLMs are usually trained with large-scale data, and many LLMs are potentially contaminated even though some of them carefully handled this problem. How to fairly compare the models is therefore necessary to decide which models to use.

This paper proposed a novel approach for contamination detection and contamination mitigation evaluation, which is composed of two novel components. The first component is **L**ength **N**ormalized **E**ntropy (LNE) for contamination detection. We are the first to apply this method to detect data contamination and surprisingly, we found it better than previous methods. On this basis, LNE-Blocking, applying LNE to adjust the intensity of blocking operation which specializes to suppressing the maximum value of output candidates during the generation process, reports a SOTA performance for contamination mitigation evaluation. Additionally, Since our method requires only two inferences for contamination mitigation evaluation, compared to 50 samplings in existing methods, TED(Dong et al., 2024), it is 25 times faster. Additionally, experiments show that our approach is highly robust across different tasks and models with varying levels of contamination.

To this end, we make three key contributions:

- We propose a novel approach for contamination detection and fairly assessing possibly contaminated models.
- Extensive experiments conducted across two tasks and multiple LLMs have yielded state-of-the-art (SOTA) results, providing strong evidence for the effectiveness of our proposed method.
- Our proposed SOTA method is 25x faster and more robust than the previous method.

## 2 MOTIVATION

Contaminated models often exhibit a high lexical overlap between their output and the ground truth, which is indicative of memory phenomena (Magar & Schwartz, 2022). As shown in Figure 1, as the degree of contamination increases, the overlap between the output generated by greedy decoding and the ground truth significantly rises. This behaviour reflects the model's tendency to memorize the training data rather than generalizing it. One solution (Dong et al., 2024) to assess the true performance of such contaminated models involves generating diverse outputs through multiple sampling and filtering out any instances of the standard ground truth. Then, they expect to derive answers that stem from the model's generalization abilities, rather than its memorized knowledge.

However, answers based on memory phenomena tend to have a very high likelihood within the model, meaning that obtaining non-memorized, generalized answers requires a large number of samplings. Previous work (Dong et al., 2024) has found that at least 50 samples are necessary to achieve satisfactory performance estimates. This process is both highly random and time-consuming.

The randomness makes it difficult to consistently generate non-memorized answers, particularly for models with heavy contamination, where such outputs are rarely sampled. This often leads to unreliable performance estimates, as we demonstrate in subsequent experiments in section 6.2.2.

Similar to how humans can rephrase their thoughts when interrupted, LLMs have the potential to generate alternative answers when their default response is blocked (Lu & Lam, 2023; Wang & Zhou, 2024). By leveraging this ability, we can apply a strategy that suppresses the generation of the token with the highest probability, referred to as blocking, without significantly impacting performance. This approach reduces the reliance on memorized content and encourages the model to draw on its generalization capabilities, as demonstrated by the reduction in ROUGE-L (Lin, 2004) similarity between the model's output and the memorized answers, as in Figure 1. Then, blocking enables the model to produce more diverse and generalized outputs, without sacrificing quality.

For example, the output of the contaminated model is *return len(set(string.lower()))*. If the model is good at coding, after applying the blocking strategy, the model could generate an equivalent piece of code using its generalization ability, such as *return len(character.lower() for character in string)*.

Additionally, as shown in Figure 1, models with varying degrees of contamination exhibit different levels of memorization, and the impact of the blocking operation varies accordingly. To effectively

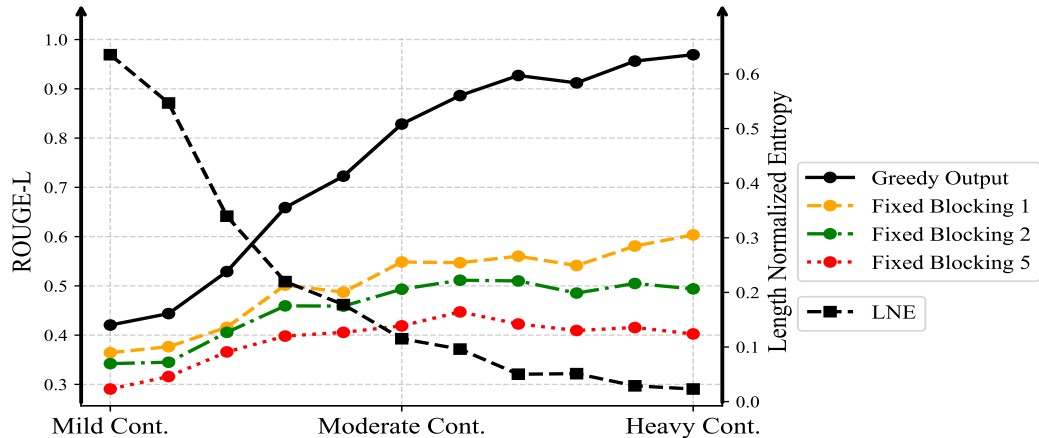

Figure 1: An example demonstrating the changes in Length-Normalized Entropy (LNE), and the impact of the blocking strategy, on model memory phenomena (ROUGE-L), in models with varying levels of contamination, where "Cont." is an abbreviation of contamination.

disrupt memorized responses, models with differing contamination levels require distinct degrees of blocking intensity. In particular, as the level of contamination increases, the blocking intensity must be intensified to counteract the effects of memory.

A natural idea, then, is to first detect the degree of contamination to better tailor the blocking strategy to ensure it disrupts memorization without negatively affecting performance. This approach allows for a more targeted and adaptive application of blocking across different contamination levels. Once the degree of contamination is detected, this information can be used to adjust the blocking intensity. Specifically, we hypothesize that a heavily contaminated model will exhibit greater certainty in its token predictions, resulting in generated text that is closer to the ground truth. This leads to lower entropy in the probability distribution at each position, as the model becomes more confident in generating memorized content. Based on this hypothesis, we propose using **L**ength **N**ormalized **E**ntropy (LNE) as a novel method to detect contamination. As shown in Figure 1, as the degree of contamination increases, the generated text becomes closer to the ground truth, resulting in higher lexical overlap and a corresponding increase in LNE. Our experiments demonstrate that this entropy-based detection method outperforms existing strategies such as perplexity in Table 4, thus we use LNE to adjust the intensity of the blocking.

## 3 RELATED WORK

**Data Contamination Detection** The issue of data contamination in large language models (LLMs) gained attention in the context of GPT-3 Brown (2020), where the vast pre-training corpus inevitably overlapped with evaluation benchmarks. To address this, GPT-3 employed n-gram overlap detection to filter out data in the training set that conflicted with test set benchmarks. Some work (Pan et al., 2020; Zhou et al., 2023; Jacovi et al., 2023; Dodge et al., 2021) exposed the serious consequences of data contamination and urged attention to this problem. However, most LLMs released to date have not made their pre-training corpus publicly available, introducing new challenges for contamination detection.

Recent efforts have attempted to detect contamination without direct access to the pre-training data. Min-k% Prob (Shi et al., 2023) calculates the average of the smallest k% probabilities of generated tokens and flags potential contamination if this average exceeds a certain threshold. Similarly, perplexity(Li, 2023) is also used to detect contamination, assuming that leaked data tends to produce lower perplexity scores. CDD (Dong et al., 2024) detects contamination by sampling multiple times to determine the proportion of highly similar samples in the dataset. The methods mentioned above do not fully leverage all the distributional information obtained from a single inference. They merely select the maximum probability at each output position to compute the score. In contrast, our

method, LNE, utilizes entropy to exploit more information, thereby achieving better contamination detection performance.

**Contamination Mitigation Evaluation** To evaluate LLMs in the context of potential data contamination, several methodologies generate new datasets that do not overlap with the model's training data, adopting a dataset-centric perspective. GSM-Plus(Li et al., 2024b) ensures that benchmark data is absent from the model's training set by reconstructing the original GSM8k dataset (Cobbe et al., 2021) through the introduction of perturbations. GSM1k(Zhang et al., 2024) involves the creation of a completely new dataset from scratch. It remains private and is only made publicly available at a future point in time. Furthermore, TreeEval(Li et al., 2024c) assesses model performance by dynamically querying the language model, using more powerful LLMs for the querying process.

One less explored but important problem is how to quantitatively assess the performance of models using datasets that have already been leaked. CleanEval(Zhu et al., 2023) proposed paraphrasing contaminated datasets using LLMs for evaluation purposes. TED (Dong et al., 2024) filters non-memorized samples through multiple sampling rounds, using these samples for contamination mitigation evaluation. However, these methods often require extensive preprocessing or multiple sampling iterations, making them highly time-consuming. LNE-Blocking is the first strategy that evaluates the performance of contaminated models using two inferences, without modifying the original dataset.

## 4 METHODLOGY

### 4.1 LNE FOR DATA CONTAMINATION DETECTION

Given a test data $x$, and its corresponding greedy decoded outout, $y$, we aim to detect whether this data was used to train the model $\mathcal{M}$. We calculate the length normalized entropy (LNE) using the probability distribution at each position of the output. For a query x, with an output of length $N$, $\mathbf{y} = (y_1, y_2, \ldots, y_N)$, the LNE is defined as:

$$\text{LNE(y)} = \frac{1}{N}\sum_{i=1}^{N} H\left(y_i\right) = -\frac{1}{N}\sum_{i=1}^{N}\sum_{j}^{V} p\left(y_i = j\right)\log p\left(y_i = j\right). \quad (1)$$

where $H(y_i)$ represents the entropy at the $i$-th position, $p\left(y_i = j\right)$ denotes the probability of the model generating the $j$-th token from the vocabulary Vocab at the $i$-th position, with $V$ representing the size of the token vocabulary.

Contamination on the test data can be detected by identifying the LNE score:

$$\text{Is\_Contaminated}(\mathcal{M}; x, y) = \begin{cases} \text{False} & \text{if} \quad \text{LNE}(y) > \xi \\ \text{True} & \text{if} \quad \text{LNE}(y) \leq \xi \end{cases} \quad (2)$$

where $\xi \in [0, log(V)]$, is a hyper-parameter that controls the threshold, $(x, y)$ represents the prompt and response output corresponding to the model $\mathcal{M}$.

### 4.2 BLOCKING OPERATION FOR LANGUAGE MODEL GENERATION

For disrupting memorization, we propose a blocking operation during the decoding process of LMs, which suppresses the token with the highest probability at a certain position during decoding.

Specifically, for a model $\mathcal{M}$ and a prompt $x$, the generation process using greedy decoding is:

$$gene(\mathcal{M}, x) = (y_1^{greedy}, y_2^{greedy}, ..., y_n^{greedy}) \quad (3)$$

At each position, the token with the highest probability is selected. This process is denoted as:

$$y_i^{greedy} = argmax(\mathcal{M}_{greedy}(x, y_{<i})) \quad (4)$$

where $\mathcal{M}_{greedy}(x, y_{<i}) \in \mathbb{R}^V$, represents the probability distribution of the $i$-th position predicted by the model through greedy decoding. When the blocking strategy is applied to the $i$-th position in a response of length $l$, the generation process is defined as:

$$block\_gene(\mathcal{M}, x, (i)) = (y_1^{greedy}, ..., y_{i-1}^{greedy}, y_i^{block}, y_{i+1}^{greedy}, ..., y_l^{greedy}) \quad (5)$$

During the process, each position before the $i$-th applies greedy decoding using equation 4. At the $i$-th position, the token with the highest probability from the distribution modified by the blocking operation is selected as the token for that position, as:

$$y_i^{block} = argmax(\mathcal{M}_{block}(x, y_{<i})) \tag{6}$$

To obtain $\mathcal{M}_{block}(x, y_{<i})$, the probability distribution with the maximum value suppressed, the process involves first identifying the index of the maximum value and then suppressing it:

$$\mathcal{M}_{greedy}[argmax(\mathcal{M}_{greedy}(x, y_{<i}))] = -\infty$$
$$\mathcal{M}_{block}(x, y_{<i}) := \mathcal{M}_{greedy}(x, y_{<i}) \tag{7}$$

After the generation of the $i$-th token, the subsequent tokens are generated using greedy decoding:

$$y_{i+1}^{greedy} = argmax(\mathcal{M}_{greedy}(x, y_{\leq i})) = argmax(\mathcal{M}_{greedy}(x, (..., y_{i-1}^{greedy}, y_i^{block}))) \tag{8}$$

### 4.3 LNE-BASED BLOCKING FOR CONTAMINATION MITIGATION EVALUATION

For data with different levels of contamination, varying intensities of disruptions are required during the generation process to disrupt the model's memorization, reflecting the model's original capabilities. It is necessary to control the disruption intensity during data generation according to different levels of contamination. This section will sequentially introduce how to control the blocking intensity and how to determine the blocking intensity based on the level of contamination.

#### 4.3.1 MULTI BLOCKING OPERATIONS FOR DISRUPTING MEMORIZATION

For highly contaminated data, only performing a blocking operation during generation is not sufficient to disrupt its memorization, necessitating an increase in the number of blocking operations to increase the disruption intensity. Starting from the first token, the blocking operation is applied $n$ times as defined below:

$$block\_gene(\mathcal{M}, x, (1, 2, ...n)) = (y_1^{block}, y_2^{block}...y_n^{block}, y_{n+1}^{greedy}, y_{n+2}^{greedy}, ..., y_l^{greedy}) \tag{9}$$

where $y_n^{block} = argmax(\mathcal{M}_{block}(x, (y_1^{block}, y_2^{block}, ..., y_{n-1}^{block})))$. After $y_n$, normal generation proceeds, where $y_{n+1}^{greedy} = argmax(\mathcal{M}_{greedy}(x, (y_1^{block}, y_2^{block}...y_n^{block})))$.

#### 4.3.2 DETERMINE THE DISRUPTING INTENSITY BASED ON THE LNE

The blocking intensity can be determined based on the level of contamination. For prompt $x$, the normal output $y^{greedy}$ is first obtained through greedy decoding, as equation 3, and then the blocking intensity corresponding to the sample is controlled by determining the number of blocking operations using Length Normalized Entropy (LNE).

Specifically, the count of blocking operations is defined as:

$$Cnt(y^{greedy}) = (1 - \frac{LNE(y^{greedy})}{\beta}) * Threshold\_Task \tag{10}$$

where $Threshold\_Task$ is the maximum number of the blocking operation on the $Task$. On each task, each data adjusts the maximum value based on its own LNE score to determine the corresponding number of blocking operations.

We found that setting $\beta = 2$ works the best in our experiments, and we postulate that dividing by 2 helps much by making an even distribution within the range of 0 to 1. The subtraction from 1 is implemented because the LNE score decreases as contamination intensifies. This adjustment allows for a larger scaling factor, resulting in more frequent blocking occurrences to effectively mitigate the contamination impact. The necessity of $Threshold\_Task$ arises from: Different task types have different average response lengths and require different blocking intensities to disrupt memorization. Therefore, the hyperparameter $Threshold\_Task$ is needed to adjust the number of blocking operations across different tasks. It is worth noting that this hyperparameter depends on the task but is independent of the model.

### 4.3.3 CONTAMINATION MITIGATION EVALUATION

After performing greedy generation once to obtain $(x, y^{\text{greedy}})$ using equation 3, we calculate the number of blocking operations, $Cnt(y^{\text{greedy}})$, using equation 10. Then, we perform $Cnt(y^{\text{greedy}})$ blocking operations to disrupt memorization, resulting in the output $y^{\text{eva}}$, as:

$$y^{eva} = block\_gene(\mathcal{M}, x, (1, 2, ...Cnt(y^{greedy}))) \tag{11}$$

$y^{eva}$ is then used to evaluate the model's performance. For an evaluation metric $\mathcal{E}$, $\mathcal{E}(y^{eva})$ is used instead of $\mathcal{E}(y^{greedy})$ to evaluate the model performance after the contamination mitigation.

The pseudocode of LNE-based Blocking for contamination mitigation evaluation is:

---
**Algorithm 1** The pseudocode of LNE-based Blocking
---
**Require:** LLM $\mathcal{M}$, the prompt of test data $x$, evaluation metric $\mathcal{E}$, and hyper-parameter $Threshold\_Task$.
**Ensure:** Evaluation performance $ep$.
 1: Obtain $y^{greedy}$ from $\mathcal{M}$ with the input $x$ via equation 3.
 2: Get the Length Normalized Entropy via equation 1.
 3: Determine the count of blocking operations $Cnt(y^{greedy})$ via equation 10.
 4: Get $y^{eva}$ from $\mathcal{M}$ using the LNE-based blocking via equation 11.
 5: Obtain $ep$ based on $\mathcal{E}(y^{eva})$.
 6: **return** $ep$.

---

## 5 EXPERIMENTAL SETUP

### 5.1 DATASET

**HumanEval (Chen et al., 2021)**: The HumanEval dataset released by OpenAI includes 164 programming problems with a function signature, docstring, body, and several unit tests. They were handwritten to ensure to be excluded from the training set of code generation models. And the initial publications(Touvron et al., 2023; Roziere et al., 2023; Nijkamp et al., 2022; Dubey et al., 2024) of the models - Llama 2, CodeLlama, CodeGen and Llama 3.1, employ the GSM8K dataset as a benchmark for evaluating their code generation performance. We assumed that these models have not been contaminated by the test set of HumanEval dataset.

**GSM8K (Cobbe et al., 2021)**: GSM8K (Grade School Math 8K) is a dataset of 8.5K high quality linguistically diverse grade school math word problems. The dataset was created to support the task of question answering on basic mathematical problems that require multi-step reasoning. The initial publications (Touvron et al., 2023; Dubey et al., 2024) of the models, Llama 2, Llama 3.1, employ the GSM8K dataset as a benchmark for evaluating their arithmetic reasoning capacity. Following previous work (Magar & Schwartz, 2022), we also note that these models are possibly subjected to contamination by the test set of the GSM8K dataset.

### 5.2 MODELS

For the code generation task, we chose four models, Llama 2, CodeLlama, CodeGen and Llama 3.1, each with 20 lora weights corresponding to different levels of contamination.

For Llama 2, CodeLlama, and CodeGen, the contaminated models were directly simulated using the LoRA weights provided by TED(Dong et al., 2024), which simulate data contamination by training LLMs using benchmark data, mixing the HumanEval test set and StarCoder data(Li et al., 2023b) at 1:1,000 ratio. For the recent model, Llama 3.1, we used its base version and employed a continued pretraining approach using the test set of HumanEval dataset to simulate contamination as (Dong et al., 2024). Due to the high cost of large-scale pre-training, we also use the LoRA to fine-tune Llama 3.1 for 20 epochs. This training was conducted on a single 4090 GPU, taking about 2 hours, with a learning rate of 1e-4. The training prompt template was formed by concatenating the HumanEval questions and answers.

For the task of arithmetic reasoning, we utilize the base versions of the Llama 2 and Llama 3.1 models and adopt a continued pretraining methodology using the test set of the GSM8K dataset

to emulate contamination, executing training for 20 epochs. This training process was carried out on a single 4090 GPU, which took approximately 20 hours. It is worth noting that the original model exhibited poor performance on the GSM8K with zero shots. The general evaluation pipeline employed the 8-shot prompt. As such, the training prompt template was devised by crafting 8-shot examples from the training set of the GSM8K. Additionally, the questions and answers from the test set of GSM8K were concatenated and appended at the end of the prompt.

For ease of analysis, we defined the uncontaminated model and the models from the first third of the 20-epoch contamination as mildly contaminated (Mild Cont.), the middle third as moderately contaminated (Moderate Cont.), and the final third as heavily contaminated (Heavy Cont.).

### 5.3 EVALUATION METRICS

For contamination detection, we utilize the **AUC** and **F1-score** for the evaluation (Dong et al., 2024). It's worth noting that when calculating the F1 score, we select the optimal threshold for each strategy at different contamination levels to obtain the best F1 for each strategy. This ensures a fair comparison of the F1 scores across the strategies.

For contamination mitigation evaluation, we measure the model's performance after contamination has been mitigated. Specifically, the performance metrics used for code generation and arithmetic reasoning tasks are **Pass@1** and exact match **Accuracy** (Dong et al., 2024), respectively. Additionally, we introduce a novel metric, **P**erformance **G**ap (**PG**), to evaluate the effectiveness of the contamination mitigation strategies. The PG is defined as:

$$\text{PG} = abs(\mathcal{E}(Y_{eva}) - \mathcal{E}(Y_{\mathcal{M}_{origin}})) \tag{12}$$

where $M_{origin}$ refers to the original uncontaminated model. PG quantifies how closely the performance of the model after mitigation aligns with that of the original uncontaminated model. The smaller the PG value is, the better the methods are.

## 6 RESULTS

### 6.1 DATA CONTAMINATION DETECTION

**Data Composition**: We validated the efficacy of contamination detection methods by distinguishing the outputs of the uncontaminated CodeLlama model on the HumanEval test set with outputs from CodeLlama models with varying degrees of contamination.

**Baselines**: We compared Length Normalized Entropy (**LNE**) with existing contamination detection methods, including: 1) **Perplexity**(Li, 2023): calculates the perplexity of the response generated by the model through greedy decoding. Lower perplexity indicates higher contamination levels; 2) **Min-k% Prob**(Shi et al., 2023): Computes the negative average log probability of the k% least probable tokens in the response generated by the model through greedy decoding. Smaller values of Min-k% Prob indicate higher contamination levels; and 3) **CDD**(Dong et al., 2024): After generating multiple sampled outputs based on the prompt, it calculates the proportion of samples where the edit distance is below a certain threshold. Higher CDD values indicate higher contamination levels. The definition of these existing methods is illustrated in Appendix D.1.

As shown in Table 1, LNE performs comparably to other baseline methods, and as the contamination level increases, the AUC of these methods can reach 1, approaching a perfect detector. However, for the more challenging setting of mild contamination, our model's strategy significantly outperforms other methods. For higher levels of contamination, our model falls slightly short of the best performance, yet under overall contamination detection, our strategy proves to be more effective.

It is also worth noting that the CDD strategy requires multiple samplings, and Perplexity relies on ground truth. In contrast, our LNE and Min-k methods do not require these, utilizing only the scores of the models' output, offering greater convenience.

### 6.2 CONTAMINATION MITIGATION EVALUATION

In this section, we evaluate the models' performance after applying the LNE-based blocking strategy to reduce contamination. We also employ PG to evaluate the effectiveness of LNE-based blocking

Table 1: Evaluation of various contamination detection strategies, where Overall refers to the combination of the outputs from the uncontaminated model with the outputs from models of all contamination levels, the **bold** is used to indicate the strategy that achieves the best performance under the current metric. Underline is used to highlight the strategy that significantly outperforms others with $p < 0.01$, as determined by a two-tailed t-test.

| | Mild Cont. | | Moderate Cont. | | Heavy Cont. | | Overall | |
|---|---|---|---|---|---|---|---|---|
| | F1 score | AUC | F1 score | AUC | F1 score | AUC | F1 score | AUC |
| Min-k | 0.706 | 0.717 | **0.942** | **0.978** | **0.989** | **0.999** | 0.839 | 0.906 |
| Perplexity | 0.627 | 0.728 | 0.925 | 0.972 | 0.986 | 0.998 | 0.844 | 0.907 |
| CDD | 0.648 | 0.674 | 0.771 | 0.846 | 0.904 | 0.952 | 0.746 | 0.826 |
| LNE | **0.775** | **0.758** | 0.927 | 0.968 | 0.973 | 0.997 | **0.854** | **0.914** |

and compare these with TED (Dong et al., 2024), a method for contamination mitigation evaluation based on sampling. The definition of TED is illustrated in Appendix D.2.

Contamination mitigation evaluation is conducted on two tasks: code generation and arithmetic reasoning. It is worth noting that the performance of the original model differs considerably between greedy and sampling decoding. Therefore, a single original model will present two distinct performances: one for greedy decoding and the other for sampling decoding.

### 6.2.1 CODE GENERATION

For the code generation task, the LNE-based Blocking strategy was employed and compared against the established sampling-based approach, TED, using the test set of HumanEval.

For this task, we set the $Threshold\_Task$ to 4. For the TED method, the edit distance threshold is set to 2, following the previous setting (Dong et al., 2024).

As shown in Table 2, the PG metric of LNE-Blocking after contamination mitigation remains close, denoting the LNE-based Blocking strategy enables models to achieve relatively stable performance restoration across different models and contamination levels after contamination mitigation. In contrast, the PG metric of TED diverges from the original model as contamination deepens, indicating insufficient stability in restoration. Additionally, the average PG metric of LNE-Blocking is small, denoting that after applying a contamination mitigation strategy to contaminated models, the evaluation performance approaches the performance of the original uncontaminated models.

With 25 times less evaluation time, our contamination mitigation strategy achieved SOTA on most contaminated models, except for the Llama2 model, where our method performed similarly to the TED method. Particularly in the heavily contaminated models CodeLlama and Llama 3.1, our method significantly outperforms TED, with up to a 10 percentage point lead on the PG metric. This is mainly due to the randomness of sampling, which causes the TED method to fail on heavily contaminated models, while the blocking strategy avoids it by controlling the triggering of sampling.

Furthermore, for models, CodeGen and Llama2, with lower contamination, the LNE-based Blocking strategy under-performs compared to TED. This may be because, with lower levels of contamination, multiple samplings can yield more diverse results to mitigate the influence of memorization. This suggests a potential for further optimization of sampling diversity in the LNE-blocking strategy.

### 6.2.2 ARITHMETIC REASONING

For the task of arithmetic reasoning, the LNE-based Blocking strategy was employed and compared against the established sampling-based approach, TED, using the test set of the GSM8K dataset.

For this task, we set the $Threshold\_Task$ to 7. For the TED method, since previous research did not evaluate this task (Dong et al., 2024), we conducted a search to identify the optimal threshold that is compatible with both Llama 2 and Llama 3.1, finding it to be 50.

Table 2: Contamination mitigation evaluation on Code Generation, where values outside the parentheses represent model performance, Pass@1, while those inside the parentheses represent the PG metric. **Bold** indicates the strategy with the best performance at the current contamination level, and underline highlights our proposed strategy, which significantly outperforms others under the current contamination level.

| Model | Strategy | Mild Cont. | Moderate Cont. | Heavy Cont. | Average |
|---|---|---|---|---|---|
| CodeGen | Sampling | 0.215 | 0.714 | 0.836 | 0.577 |
| | Greedy | 0.279 | 0.819 | 0.913 | 0.653 |
| | TED | 0.138 (**0.031**) | 0.234 (0.112) | 0.211 (0.072) | 0.188 (0.072) |
| | LNE-Blocking | 0.088 (0.076) | 0.113 (**0.052**) | 0.117 (**0.037**) | 0.108 (**0.056**) |
| Llama 2 | Sampling | 0.210 | 0.659 | 0.798 | 0.556 |
| | Greedy | 0.248 | 0.742 | 0.861 | 0.609 |
| | TED | 0.112 (**0.017**) | 0.128 (**0.021**) | 0.114 (0.036) | 0.114 (**0.024**) |
| | LNE-Blocking | 0.133 (0.02) | 0.134 (0.037) | 0.128 (**0.018**) | 0.132 (0.025) |
| CodeLlama | Sampling | 0.341 | 0.7 | 0.808 | 0.613 |
| | Greedy | 0.413 | 0.784 | 0.87 | 0.682 |
| | TED | 0.290 (0.078) | 0.392 (0.174) | 0.375 (0.137) | 0.345 (0.129) |
| | LNE-Blocking | 0.297 (**0.035**) | 0.282 (**0.032**) | 0.271 (**0.045**) | 0.283 (**0.037**) |
| Llama 3.1 | Sampling | 0.437 | 0.879 | 0.947 | 0.739 |
| | Greedy | 0.480 | 0.893 | 0.936 | 0.758 |
| | TED | 0.374 (0.069) | 0.257 (0.083) | 0.176 (0.169) | 0.273 (0.101) |
| | LNE-Blocking | 0.340 (**0.059**) | 0.364 (**0.038**) | 0.305 (**0.067**) | 0.333 (**0.054**) |

Table 3: Contamination mitigation evaluation on GSM8K, where values outside the parentheses represent model performance, Accuracy, while those inside the parentheses represent the PG metric.

| Model | Strategy | Mild Cont. | Moderate Cont. | Heavy Cont. | Average |
|---|---|---|---|---|---|
| Llama 2 | Sampling | 0.340 | 0.715 | 0.874 | 0.627 |
| | Greedy | 0.226 | 0.637 | 0.853 | 0.556 |
| | TED | 0.315 (0.096) | 0.278 (0.119) | 0.113 (0.162) | 0.232 (0.122) |
| | LNE-Blocking | 0.155 (**0.02**) | 0.224 (**0.079**) | 0.222 (**0.075**) | 0.198 (**0.057**) |
| Llama 3.1 | Sampling | 0.759 | 0.939 | 0.995 | 0.889 |
| | Greedy | 0.582 | 0.889 | 0.993 | 0.807 |
| | TED | 0.718 (**0.018**) | 0.306 (0.414) | 0.050 (0.694) | 0.379 (0.346) |
| | LNE-Blocking | 0.440 (0.114) | 0.487 (**0.068**) | 0.488 (**0.065**) | 0.471 (**0.084**) |

As shown in Table 3, similar to the task of code generation, our method enables models to achieve relatively stable performance restoration across different models and contamination levels after contamination mitigation. And TED suffers from insufficient stability in restoration.

Simultaneously, with 25 times less computational costs, our contamination mitigation strategy achieved SOTA on most contaminated models. Particularly in the heavily contaminated models, our method significantly outperforms TED. It is worth noting that the PG metric increases dramatically to about 0.414 and 0.694 when applying TED to the Mildly and heavily contaminated Llama

3.1, denoting it fails completely. This is also due to its sampling randomness, which prevents it from generating diverse answers when the probability of memorized answers is high.

## 6.3 Ablation study

Table 4: Ablation study of the components of LNE-based Blocking, where values outside the parentheses represent model performance, Pass@1, while those inside the parentheses represent the PG metric, and Perplexity is denoted as PPL. Fixed Blocking 1, 2, and 5 refer to the fixed number of blocking operations applied for any given prompt.

| Strategy | Mild Cont. | Moderate Cont. | Heavy Cont. | Average |
|---|---|---|---|---|
| Greedy Decoding | 0.413 | 0.784 | 0.87 | 0.682 |
| Fixed Blocking 1 | 0.345 (0.073) | 0.430 (0.119) | 0.413 (0.1) | 0.394 (0.097) |
| Fixed Blocking 2 | 0.338 (0.076) | 0.372 (0.07) | 0.351 (**0.033**) | 0.352 (0.062) |
| Fixed Blocking 3 | 0.309 (0.05) | 0.305 (0.037) | 0.288 (0.035) | 0.304 (0.041) |
| Fixed Blocking 4 | 0.241 (0.085) | 0.274 (0.04) | 0.271 (0.043) | 0.261 (0.057) |
| PPL-Blocking | 0.300 (0.044) | 0.280 (0.037) | 0.274 (0.047) | 0.284 (0.042) |
| LNE-Blocking | 0.297 (**0.035**) | 0.282 (**0.032**) | 0.271 (0.045) | 0.283 (**0.037**) |

In this section, we analyze the effects of each component in LNE-based Blocking.

As shown in Table 4, when using a fixed number of blocking steps to restore the performance of models with varying levels of contamination, the degree of restoration differs. With fewer blocking steps, the performance of models with mild contamination can be well restored. However, as the number of blocking steps increases, models with more severe contamination are restored more effectively, while the restoration of less contaminated models becomes less optimal. This demonstrates the effectiveness of employing a contamination detection strategy to adjust the blocking intensity according to the contamination level.

When using the Perplexity (PPL) instead of LNE, as shown in Table4, its performance recovery after contamination mitigation is less effective than LNE for models with mild contamination. This indicates that PPL's adjustment of blocking intensity is less effective compared to LNE, as PPL is less proficient in detecting contamination in models with lower levels of contamination, as illustrated in Table 1. The significant effectiveness of LNE in cases of low contamination leads to the best average performance, further validating our choice of LNE as a method for adjusting blocking intensity.

## 7 Conclusion

In this paper, we introduced LNE-Blocking, a novel approach for contamination detection and contamination mitigation evaluation. For the first component, Length Normalized Entropy (LNE) is used for contamination detection. On this basis, LNE-Blocking is used for contamination mitigation evaluation by applying LNE to adjust the intensity of the blocking operation. Our extensive experiments on code generation and mathematical reasoning tasks demonstrate that LNE-Blocking consistently achieves state-of-the-art performance while being significantly more efficient, reducing computational costs by 25x compared to previous methods, and more robust than the previous method across tasks and models with varying levels of contamination.

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

# A  DATA CONTAMINATION DETECTION ON OTHER MODELS

We validated the efficacy of contamination detection methods by distinguishing the outputs of the uncontaminated models (Llama 2, Llama 3.1, and CodeGen) on the HumanEval test set from their respective outputs with varying degrees of contamination.

As shown in table 5, LNE achieves state-of-the-art performance across nearly all contamination levels for both Llama2 and Llama3.1 models. In the contamination detection results for the Code-Gen models, LNE performs only worse than CDD and outperforms other contamination detection methods based on single inference. It is worth noting that CDD requires 50 sampling iterations for contamination detection, while our method operates with just a single inference.

Table 5: Evaluation of contamination detection strategies across different contamination levels of three models: Llama 2, Llama 3.1, and CodeGen.

| Model | Strategy | Mild Cont. | | Moderate Cont. | | Heavy Cont. | | Overall | |
|---|---|---|---|---|---|---|---|---|---|
| | | F1 score | AUC | F1 score | AUC | F1 score | AUC | F1 score | AUC |
| CodeGen | Min-k | 0.736 | 0.700 | 0.833 | 0.892 | 0.878 | 0.934 | 0.778 | 0.847 |
| | Perplexity | 0.732 | 0.710 | 0.833 | 0.895 | 0.907 | 0.949 | 0.771 | 0.856 |
| | CDD | 0.686 | 0.707 | **0.867** | **0.915** | **0.963** | **0.992** | **0.813** | **0.876** |
| | LNE | **0.741** | **0.725** | 0.827 | 0.905 | 0.914 | 0.958 | 0.777 | 0.867 |
| Llama 2 | Min-k | 0.701 | 0.579 | 0.859 | 0.890 | 0.921 | 0.967 | 0.774 | 0.820 |
| | Perplexity | 0.727 | 0.635 | 0.875 | 0.928 | 0.940 | 0.984 | 0.788 | 0.857 |
| | CDD | 0.712 | **0.742** | 0.824 | 0.887 | 0.935 | 0.972 | 0.798 | **0.869** |
| | LNE | **0.738** | 0.648 | **0.884** | **0.932** | **0.942** | **0.986** | **0.800** | 0.863 |
| Llama 3.1 | Min-k | 0.728 | 0.681 | 0.930 | 0.967 | 0.962 | 0.992 | 0.830 | 0.889 |
| | Perplexity | **0.745** | **0.709** | 0.944 | 0.983 | 0.972 | 0.997 | 0.839 | **0.905** |
| | CDD | 0.667 | 0.608 | 0.891 | 0.936 | 0.959 | 0.988 | 0.805 | 0.853 |
| | LNE | 0.744 | 0.707 | **0.952** | **0.985** | **0.979** | **0.998** | **0.844** | **0.905** |

# B  CONTAMINATION MITIGATION EVALUATION ON A NEWLY RELEASED DATASET

Relying solely on the declarations from the model developers may not guarantee that the model has not been contaminated by the test dataset, which weakens the validity of our prior contamination simulation and may affect the experimental conclusions. Therefore, it is essential to ensure that the test dataset was not included in the training set of the uncontaminated model in order to validate the effectiveness of our strategy.

To address this issue, we have used a new dataset, GSM-Plus (Li et al., 2024a), to simulate contamination. It is an augmented version of GSM8K with various mathematical perturbations, including numerical variation, arithmetic variation, problem understanding challenges, distractor insertion, and critical thinking tasks. This dataset was released in January 2024, which is after the release of Llama2. Given the randomness of these perturbations and the innovative nature of the techniques employed, it is highly likely that the original uncontaminated version of Llama2 was not exposed to contamination during its training.

As shown in Table 6, similar to the dataset mentioned earlier, our method enables models to achieve relatively stable performance restoration across different models and contamination levels after contamination mitigation. In contrast, TED suffers from insufficient stability in restoration. It is worth

noting that the PG metric increases dramatically to about 0.44 when applying TED to the heavily contaminated Llama 3.1, indicating a complete failure.

For the Llama 2, whose origin model is less prone to contamination risks, our method achieves a performance restoration with a maximum error of only 6% across all contamination levels.

Therefore, when the original model is less likely to be contaminated by the test dataset, LNE-Blocking remains effective while being 25 times faster across most contaminated models.

Table 6: Contamination mitigation evaluation on GSM-Plus, where values outside the parentheses represent model performance, Accuracy, while those inside the parentheses represent the PG metric.

| Model | Strategy | Mild Cont. | Moderate Cont. | Heavy Cont. | Average |
|---|---|---|---|---|---|
| Llama 2 | Sampling | 0.254 | 0.690 | 0.754 | 0.550 |
| | Greedy | 0.169 | 0.726 | 0.813 | 0.547 |
| | TED | 0.249 (0.101) | 0.269 (0.121) | 0.174 (**0.014**) | 0.232 (0.085) |
| | LNE-Blocking | 0.123 (**0.027**) | 0.16 (**0.06**) | 0.144 (0.048) | 0.143 (**0.045**) |
| Llama 3.1 | Sampling | 0.685 | 0.943 | 0.981 | 0.861 |
| | Greedy | 0.518 | 0.850 | 0.899 | 0.744 |
| | TED | 0.67 (0.113) | 0.357 (0.269) | 0.139 (0.44) | 0.406 (0.259) |
| | LNE-Blocking | 0.338 (**0.11**) | 0.336 (**0.112**) | 0.33 (**0.113**) | 0.336 (**0.111**) |

## C  ANALYSIS OF BLOCKING TOKENS

In this section, we illustrate the token changes that occur during the execution of the blocking operation. We pair the tokens selected for blocking with their respective alternative tokens and visualize the top 20 token with the highest frequencies in figure 2.

Furthermore, we provide examples in table 7, displaying the complete model outputs both before and after applying the blocking operation, to demonstrate its impact on the model's response.

## D  FORMULAS AND PRINCIPLES FOR COMPARISON METHODS

### D.1  DATA CONTAMINATION DETECTION

Consider the output of the model's greedy decoding as $y$. Below are the definitions of several contamination detection methods.

**Perplexity**:calculates the perplexity of the response generated by the model through greedy decoding, as:

$$\text{Perplexity} = \exp\left(-\frac{1}{N}\sum_{i=1}^{N}\log P(y_i)\right) \tag{13}$$

where N is the length of the greedy decoded output, $y$. Lower perplexity indicates higher contamination levels.

**Mink% Prob**: Computes the negative average log probability of the k% least probable tokens in the response generated by the model through greedy decoding, as:

$$\text{Min-k}(y) = -\frac{1}{E}\sum_{y_i \in \text{Min-k\%(y)}}\log P(y_i) \tag{14}$$

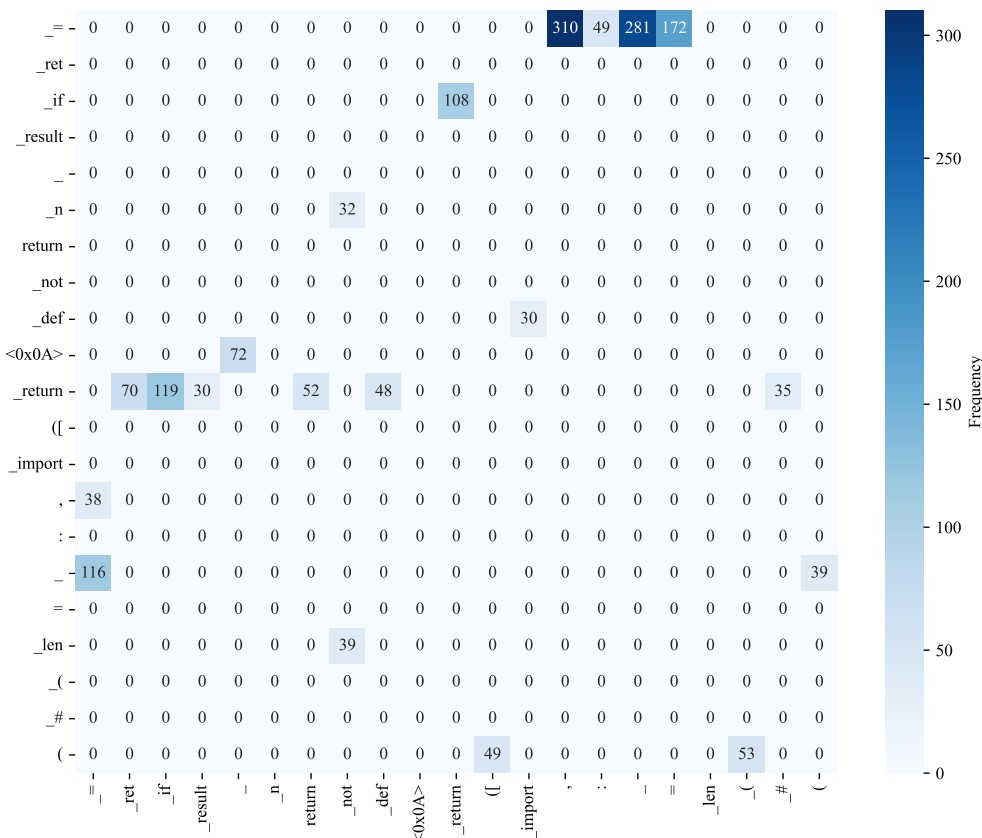

Figure 2: A heatmap of token replacement frequency during the blocking operation.

where E is the size of the Min-K%(y) set, and Min-K%(y) is the set of k% least probable tokens in the response. Smaller values of Min-k% Prob indicate higher contamination levels.

**CDD**: After generating multiple sampled outputs based on the prompt, it calculates the proportion of samples where the edit distance is below a certain threshold, as:

$$\text{CDD} = \frac{1}{M} \sum_{m=1}^{M} \mathbf{1} \left( \text{EditDist}(y^m, y) \leq \delta \right) \tag{15}$$

where $y_m$ is the $m$-th sampled output, $\text{EditDist}(y_m, y)$ is the edit distance between the sampled output and the greedy decoded output, and $\delta$ is a predefined threshold for the edit distance. $M$ is the total number of sampled outputs. Higher CDD values indicate higher contamination levels.

## D.2 CONTAMINATION MITIGATION EVALUATION

TED: filters non-memorized samples through multiple sampling rounds, using these samples for contamination mitigation evaluation. It is illustrated as:

$$S_e = \left\{ s \mid s \in S \wedge \text{EditDist}\left(s, s^{greedy}\right) > \tau \right\} \tag{16}$$

where $S$ is the set of outputs sampled from LLM, $s^{greedy}$ denotes the output generated by the LLM using greedy decoding, and $\tau$ is a predefined threshold for the edit distance. The set $S_e$ contains only those samples from $S$ for which the edit distance to the greedy decoding output exceeds the threshold $\tau$.

Table 7: Examples of LLM Outputs Before and After Blocking, where Replaced Pairs (A, B) indicates the replacement of token A with candidate token B during blocking.

| Replaced Pairs | Origin Outputs |
|---|---|
| | Outputs after Blocking |
| (_if , _return) | if not numbers:\n return 0.0\n mean = sum(numbers) / len(numbers)\n return sum(abs(x - mean) for x in numbers) / len(numbers) |
| | return float((sum(abs(x - mean(numbers)) for x in numbers)) / len(numbers)) |
| (_return, _ret) | return [x + 1 for x in l] |
| | ret_l=[]\n for i in l:\n ret_l.append(i+1)\n return ret_l |
| (_return, return) | return number % 1.0 |
| | returnnumber=number%1\n return returnnumber |
| (_return, _if) | return ''.join(map(lambda x, y: str(int(x) înt(y)), a, b)) |
| | if a != b:\n raise ValueError("Inputs must be equal")\n return "".join(map(lambda x, y: str(int(x) înt(y)), a, b)) |
| (_return, _#) | return ' '.join(str(i) for i in range(n + 1)) |
| | # TODO\n return ' '.join(str(i) for i in range(n + 1)) |
| (_return, _def) | return [len(p) for p in paren_string.split() if p] |
| | def parse(s):\n if s[0] == '(':\n return 1 + parse(s[1:])\n else:\n return 0\n \n return [parse(s) for s in paren_string.split()] |