# OpenReview forum: "LBG: LNE-based Blocking Generation Against Data Contamination on Large Language Models"
_ICLR.cc/2025/Conference — Submitted to ICLR 2025_

### Official Review · Reviewer_D5E5 · 2024-10-29

**Soundness:** 2
**Presentation:** 2
**Contribution:** 2
**Rating:** 3
**Confidence:** 4

**Summary:**

Benchmark leakage will significantly impact a fair evaluation of the LLM ability. In this paper, the authors employ Length Normalized Entropy (LNE) to detect data contamination and achieve contamination mitigation evaluation. For the detection, the intuition is that memorized output may share an overlap with the ground truth. By observing the entropy pattern of the LLM on the ground truth, this work detects contaminated samples. This work also proposes a LNE-based method to block the generation of memorized samples.The results show that the methods can achieve excellent detection and mitigation results compared with baselines.

**Strengths:**

1. **Focus on important problems.** As more and more diverse benchmarks and LLMs are released, the contamination will mislead our understanding of the LLM progress. I acknowledge that both contamination detection and contamination-free evaluation are crucial research problems.
2. **Intriguing motivation**: I admire the motivation discussion in Section 2.

**Weaknesses:**

1. **Unclear effects of LNE blocking.** Even if the exactly memorized ground truth can be blocked, the impacts of the leaked data points on the LLM performance are not entirely removed. The alternate answers, as mentioned in Section 2, will be inspired by the leaked data points. It is a little strange that the performance after LNE-Blocking will become close to the original uncontaminated model.
2. **Crucial experiments are missing.** As we can never forecast the contamination behaviors, we may expect to employ contamination mitigation evaluation for all LLMs. What about the original model performance or any other uncontaminated LLM performance under LTE blocking?
3. **Realistic testbeds should be evaluated.** I suggest collecting realistic benchmark contamination reported by the LLM development teams instead of focusing on the manually contaminated LLMs. To be honest, training LoRA adapter for 20 epochs will result in a total memorization of the gold truths. The contaminated models are weak subjects for the task of contamination detection. For real cases, the leaked tokens will be mixed up with other riskless tokens and then be trained for a limited number of epochs. In such a case, it definitely leads to a higher chance of false positives for both contamination detection and contamination mitigation evaluation.
4. **Writing quality requires improvements.**
- In Eq.1, I guess $N$ is equal to $l$, both corresponding to the output length.
- In Eq. 12, what about the meaning of $M_{original}$? The authors should define it.
- The setting for $\beta$ in line 229 is a little beyond comprehension. The postulation is not convincing and requires the support of at least empirical results. Even so, it is still unclear why an even distribution within the range of 0 to 1 is desirable.
- After walking through the whole paper, I am still confused about the identity of $y$ in Eq. 1. Does it correspond to the generated output by the LLM or the ground truth?

**Questions:**

- The LTE blocking is operated at the beginning of the generation. What's the rationale behind the choice? And how does it impact the contamination mitigation evaluation?
- Self-consistency is a widely adopted strategy for evaluating LLM abilities. Can you demonstrate how to integrate LTE blocking with self-consistency?

---

> ### Author Response · Authors · 2024-12-02
> **Weakness one**
>
> Thank you sincerely for your thorough reading of our work and detailed
> suggestions for improvements. We are glad that you recognize the
> significance of the problems and appreciate the motivation discussed in
> Section 2.
>
> We would like to address your concerns in the following.
>
> ## Weakness one:
>
> Our approach aims to restore the performance of the contaminated model
> to that of the uncontaminated model by interrupting the model’s "ground
> truth" memory. As you rightly pointed out, we cannot fully eliminate the
> influence of the memory content on the candidate tokens after performing
> the blocking operation. However, we believe that the impact of data
> leakage on the candidate tokens is much smaller than its impact on the
> greedy decoded tokens. The fact that LNE-Blocking is effective in
> recovering the performance of the contaminated model to be closer to the
> uncontaminated model supports this argument.
>
> Furthermore, to the best of our knowledge, we are the first to explore
> restoring the performance of contaminated models based on logits. More
> mechanistic and detailed studies on this topic could be the focus of
> future research.

---

> > ### Author Response · Authors · 2024-12-02
> > **Weakness two**
> >
> > ## Weakness two:
> >
> > Thank you for raising this important point. We completely agree that it
> > is crucial to evaluate the performance of the uncontaminated model under
> > LNE-blocking. Initially, we grouped the unpolluted model with the mildly
> > contaminated model, but we have now separated them and presented their
> > individual performances more clearly. This includes evaluations on tasks
> > such as code generation and arithmetic reasoning, as shown in the
> > updated table below.
> > Although our method does not outperform TED on the uncontaminated model,
> > we would like to emphasize that our approach has not failed. It still
> > provides valuable insights for non-sampling-based contamination
> > mitigation tasks, which we believe will contribute to future
> > advancements in this area.
> >
> > | Model     | Strategy     | Uncont.       | Mild Cont.    | Moderate Cont. | Heavy Cont.   | Average        |
> > |-----------|--------------|---------------|---------------|----------------|---------------|----------------|
> > | CodeGen   | Sampling     | 0.122         | 0.246         | 0.714          | 0.836         | 0.577          |
> > | CodeGen   | Greedy       | 0.165         | 0.317         | 0.819          | 0.913         | 0.653          |
> > | CodeGen   | TED          | 0.106 (**0.016**) | 0.148 (**0.036**) | 0.234 (0.112)  | 0.211 (0.072) | 0.188 (0.072)  |
> > | CodeGen   | LNE-Blocking | 0.073 (0.091) | 0.094 (0.071) | 0.113 (**0.052**)  | 0.117 (**0.037**) | 0.108 (**0.056**)  |
> > | Llama 2   | Sampling     | 0.111         | 0.243         | 0.659          | 0.798         | 0.556          |
> > | Llama 2   | Greedy       | 0.128         | 0.289         | 0.742          | 0.861         | 0.609          |
> > | Llama 2   | TED          | 0.095 (**0.016**) | 0.118 (0.017) | 0.128 (**0.021**)  | 0.114 (0.036) | 0.114 (**0.024**)  |
> > | Llama 2   | LNE-Blocking | 0.098 (0.03)  | 0.144 (**0.016**) | 0.134 (0.037)  | 0.128 (**0.018**) | 0.132 (0.025)  |
> > | CodeLlama | Sampling     | 0.218         | 0.382         | 0.7            | 0.808         | 0.613          |
> > | CodeLlama | Greedy       | 0.311         | 0.447         | 0.784          | 0.87          | 0.682          |
> > | CodeLlama | TED          | 0.205 (**0.013**) | 0.318 (0.099) | 0.392 (0.174)  | 0.375 (0.137) | 0.345 (0.129)  |
> > | CodeLlama | LNE-Blocking | 0.268 (0.043) | 0.307 (**0.033**) | 0.282 (**0.032**)  | 0.271 (**0.045**) | 0.283 (**0.037**)  |
> > | Llama 3.1 | Sampling     | 0.329         | 0.474         | 0.879          | 0.947         | 0.739          |
> > | Llama 3.1 | Greedy       | 0.348         | 0.524         | 0.893          | 0.936         | 0.758          |
> > | Llama 3.1 | TED          | 0.306 (**0.023**) | 0.397 (0.084) | 0.257 (0.083)  | 0.176 (0.169) | 0.273 (0.101)  |
> > | Llama 3.1 | LNE-Blocking | 0.293 (0.055) | 0.356 (**0.061**) | 0.364 (**0.038**)  | 0.305 (**0.067**) | 0.333 (**0.054**)  |
> >
> > | Model     | Strategy     | Uncont.       | Mild Cont.    | Moderate Cont. | Heavy Cont.   | Average        |
> > |-----------|--------------|---------------|---------------|----------------|---------------|----------------|
> > | Llama 2   | Sampling     | 0.221         | 0.38          | 0.715          | 0.874         | 0.627          |
> > | Llama 2   | Greedy       | 0.145         | 0.252         | 0.637          | 0.853         | 0.556          |
> > | Llama 2   | TED          | 0.217 (**0.005**) | 0.348 (0.126) | 0.278 (0.119)  | 0.113 (0.162) | 0.232 (0.122)  |
> > | Llama 2   | LNE-Blocking | 0.133 (0.012) | 0.163 (**0.023**) | 0.224 (**0.079**)  | 0.222 (**0.075**) | 0.198 (**0.057**)  |
> > | Llama 3.1 | Sampling     | 0.719         | 0.772         | 0.939          | 0.995         | 0.889          |
> > | Llama 3.1 | Greedy       | 0.555         | 0.592         | 0.889          | 0.993         | 0.807          |
> > | Llama 3.1 | TED          | 0.704 (**0.016**) | 0.723 (**0.019**) | 0.306 (0.414)  | 0.05 (0.694)  | 0.379 (0.346)  |
> > | Llama 3.1 | LNE-Blocking | 0.475 (0.08)  | 0.429 (0.126) | 0.487 (**0.068**)  | 0.488 (**0.065**) | 0.471 (**0.084**)  |

---

> > > ### Author Response · Authors · 2024-12-02
> > > **Weakness three**
> > >
> > > ## Weakness three:
> > >
> > > 1.  Thank you for your insightful comment. We apologize for not clearly
> > >     explaining the contamination data mixing strategy in our original
> > >     submission. In our experiments, we made an effort to simulate
> > >     contamination as realistically as possible. In the code generation
> > >     task, the LoRA weights were provided by TED, with contamination
> > >     simulated by mixing the HumanEval test set (containing risk-laden
> > >     tokens) and StarCoder data[1] (treated as riskless tokens) at a 1:1,000 ratio. By
> > >     mixing additional data into the contaminated dataset, we reduce the
> > >     likelihood that the model will completely memorize the reference
> > >     answers. We have now provided a detailed explanation of this
> > >     contamination setup in the revised version of the paper.
> > >
> > > 2.  As you pointed out, the contamination detection task becomes easier
> > >     as the level of contamination increases, even though we mix riskless
> > >     tokens into the contamination process, as shown in the contamination
> > >     detection results in the paper. However, for contamination
> > >     mitigation evaluation, as the contamination level intensifies,
> > >     sampling-based methods tend to lose effectiveness. In contrast,
> > >     blocking-based strategies can recover the original model performance
> > >     even under heavy contamination. This approach avoids scenarios where
> > >     heavy contamination would lead to the abandonment of evaluation,
> > >     thereby making full use of the existing benchmarks.
> > >
> > > 3.  We would also like to share a set of data about contamination
> > >     mitigation evalution in order to address any concerns you may
> > >     have.We have introduced a new dataset, GSM-Plus[2], an augmented version of
> > >     GSM8K, which incorporates various mathematical perturbations such as
> > >     numerical variation, arithmetic variation, challenges in problem
> > >     understanding, distractor insertion, and critical thinking tasks.
> > >     This dataset was released in January 2024, after the release of
> > >     Llama2. Given the randomness of these perturbations and the
> > >     innovative nature of the techniques employed, it is highly likely
> > >     that the original uncontaminated version of Llama2 was not exposed
> > >     to contamination during its training.
> > >
> > >     We have further validated the effectiveness of the LNE-Blocking
> > >     strategy on this new dataset, as shown in the table below. We have also
> > >     included this additional information in the appendix of the revised
> > >     paper.
> > >
> > >     | Model     | Strategy     | Uncont.           | Mild Cont.    | Moderate Cont. | Heavy Cont.   | Average        |
> > >     |-----------|--------------|-------------------|---------------|----------------|---------------|----------------|
> > >     | Llama 2   | Sampling     | 0.151             | 0.274         | 0.683          | 0.752         | 0.542          |
> > >     | Llama 2   | Greedy       | 0.090             | 0.208         | 0.651          | 0.747         | 0.505          |
> > >     | Llama 2   | TED          | 0.152 (**0.001**) | 0.262 (0.111) | 0.305 (0.154)  | 0.143 (**0.008**) | 0.235 (0.089)  |
> > >     | Llama 2   | LNE-Blocking | 0.096 (0.007)     | 0.117 (**0.028**) | 0.14 (**0.051**)   | 0.139 (0.049) | 0.13 (**0.04**)    |
> > >     | Llama 3.1 | Sampling     | 0.586             | 0.689         | 0.947          | 0.982         | 0.853          |
> > >     | Llama 3.1 | Greedy       | 0.426             | 0.542         | 0.917          | 0.982         | 0.788          |
> > >     | Llama 3.1 | TED          | 0.586 (**0.000**) | 0.677 (0.091) | 0.378 (0.275)  | 0.119 (0.467) | 0.408 (0.252)  |
> > >     | Llama 3.1 | LNE-Blocking | 0.35 (0.076)      | 0.338 (**0.088**) | 0.315 (**0.111**)  | 0.327 (**0.099**) | 0.328 (**0.098**)  |

---

> > > > ### Author Response · Authors · 2024-12-02
> > > > **Weakness four**
> > > >
> > > > ## Weakness four:
> > > >
> > > > Thank you for your thoughtful and detailed feedback. We have made the
> > > > following revisions in response to your comments about writing quality:
> > > >
> > > > 1.  As you pointed out, *l* and *N* were used interchangeably to
> > > >     indicate the output length, which caused confusion.
> > > >
> > > > 2.   $M_{origin}$ refers to the original
> > > >     uncontaminated model.
> > > >
> > > > 3.  1.  The reason why an even distribution within the range of 0 to 1
> > > >         is desirable is due to the interpretability and consistency of
> > > >         the Threshold\_Task in relation to the blocking intensity.
> > > >
> > > >         We define the LNE normalized by *β* as the control factor. For
> > > >         each sample, the control factor, based on contamination level,
> > > >         is multiplied by Threshold\_Task to determine blocking
> > > >         intensity. When the control factor is between 0 and 1,
> > > >         Threshold\_Task reflects the maximum blocking intensity, making
> > > >         it easier to interpret.
> > > >
> > > >         If the control factor exceeds this range, it still scales
> > > >         Threshold\_Task, but the meaning of Threshold\_Task as the
> > > >         maximum blocking count becomes unclear. It then acts merely as a
> > > >         scaling factor, reducing the interpretability of how
> > > >         Threshold\_Task relates to the blocking behavior.
> > > >
> > > >         Thus, keeping the control factor within 0 to 1 ensures that
> > > >         Threshold\_Task remains a clear upper limit for blocking
> > > >         intensity, allowing for consistent and intuitive adjustments
> > > >         based on contamination.
> > > >
> > > >      2. We set *β* to 2 primarily because we observed that, for
> > > >       different models, the range of LNE generally falls between 0
> > > >       and 2. Therefore, we normalized it by dividing by *β* = 2.
> > > >       Furthermore, we conducted additional experiments and found that
> > > >       *β* = 2 indeed yields the optimal results. The experimental
> > > >       results are as follows:
> > > >
> > > >        | Strategy          | Uncont.       | Mild Cont.    | Moderate Cont. | Heavy Cont.   | Average        |
> > > >        |-------------------|---------------|---------------|----------------|---------------|----------------|
> > > >        | Greedy            | 0.311         | 0.447         | 0.784          | 0.87          | 0.682          |
> > > >        | LNE-Blocking(β=1) | 0.287 (**0.024**) | 0.327 (0.053) | 0.282 (**0.032**)  | 0.265 (0.047) | 0.29 (0.041)   |
> > > >        | LNE-Blocking(β=2) | 0.268 (0.043) | 0.307 (**0.033**) | 0.282 (**0.032**)  | 0.271 (**0.045**) | 0.283 (**0.037**)  |
> > > >        | LNE-Blocking(β=3) | 0.220 (0.092) | 0.289 (0.039) | 0.281 (0.034)  | 0.268 (0.049) | 0.272 (0.044)  |
> > > >
> > > >       We will provide a more detailed explanation of these reasons in
> > > >       the later version of the paper.
> > > >
> > > > 4.  In Eq. 1, *y* refers to the greedy decoding output generated by the
> > > >     model, not the ground truth. Both LNE and LNE-Blocking do not rely
> > > >     on the reference answer during their operation. And the reference
> > > >     answer is only used in the final performance evaluation for
> > > >     comparison purposes.
> > > >
> > > > We apologize for these oversights and have corrected them in the updated
> > > > version.

---

> > > > > ### Author Response · Authors · 2024-12-02
> > > > > **Question one and two**
> > > > >
> > > > > ## Question one:
> > > > >
> > > > > 1.  "The LNE-blocking is operated at the beginning of the generation.
> > > > >     What’s the rationale behind the choice? "
> > > > >
> > > > >     If the blocking operation occurs earlier in the generation process,
> > > > >     it blocks the most likely memorized tokens while selecting candidate
> > > > >     tokens. This allows the LLM to use the previously generated
> > > > >     candidate tokens to adjust the response logic easlier when
> > > > >     generating subsequent tokens, demonstrating its generalization
> > > > >     ability (which corresponds to the performance of the uncontaminated
> > > > >     LLM). A study has shown that sampling the first token can trigger
> > > > >     the model’s chain of thought ability. This may offer an alternative
> > > > >     explanation for why early interruption of the model’s generation of
> > > > >     memorized content enables it to leverage its generalization ability
> > > > >     to rephrase the response[3]. In contrast, if blocking happens later, the LLM
> > > > >     continues to rely on the pre-determined logic of the previously
> > > > >     generated tokens, which is influenced by the contaminated data,
> > > > >     leading to performance of contamination mitigated LLM consistent
> > > > >     with the contaminated LLM.
> > > > >
> > > > > 2.  "And how does it impact the contamination mitigation evaluation?"
> > > > >
> > > > >     We have included a heatmap of token replacement frequency during the
> > > > >     blocking operation in the Appendix C of the revised paper. Below are some examples
> > > > >     for your reference.
> > > > >
> > > > >     Replace _if with _return:
> > > > >     ```python
> > > > >     if not numbers:
> > > > >         return 0.0
> > > > >     mean = sum(numbers) / len(numbers)
> > > > >     return sum(abs(x - mean) for x in numbers) / len(numbers)
> > > > >
> > > > >     ↓↓↓
> > > > >
> > > > >     return float((sum(abs(x - mean(numbers)) for x in numbers)) / len(numbers))
> > > > >     ```
> > > > >     Replace _return with _ret:
> > > > >     ```python
> > > > >     return [x + 1 for x in l]
> > > > >
> > > > >     ↓↓↓
> > > > >
> > > > >     ret_l=[]
> > > > >     for i in l:
> > > > >         ret_l.append(i+1)
> > > > >     return ret_l
> > > > >     ```
> > > > >     Replace _return with return:
> > > > >     ```python
> > > > >     return number % 1.0
> > > > >
> > > > >     ↓↓↓
> > > > >
> > > > >     returnnumber=number%1
> > > > >     return returnnumber
> > > > >     ```
> > > > >
> > > > > ## Question two:
> > > > >
> > > > > Self-consistency is a common strategy for evaluating LLM performance,
> > > > > involving multiple samples to assess reliability. However, integrating
> > > > > LNE-blocking with self-consistency presents challenges.
> > > > >
> > > > > In self-consistency, multiple samples are generated, and the most
> > > > > consistent output is selected. Some of these samples may not be outputs
> > > > > generated by the model based on memory. LNE-blocking, on the other hand,
> > > > > requires calculating the level of contamination after each sample was generated and
> > > > > adjusting the blocking intensity accordingly. Since sampling is random,
> > > > > the blocking intensity determined for one sample may not apply to
> > > > > others, making direct integration difficult.
> > > > >
> > > > > To combine LNE-blocking with self-consistency, further research is
> > > > > needed to develop an independent contamination assessment strategy. This
> > > > > would ensure accurate blocking intensity for each sample and allow for
> > > > > accurate contamination mitigation evaluation within the self-consistency
> > > > > framework.

---

> > > > > > ### Author Response · Authors · 2024-12-02
> > > > > > **Novelty Beyond Technique**
> > > > > >
> > > > > > ## Novelty Beyond Technique
> > > > > > **Last but importantly**, we would like to emphasize the novelty of our
> > > > > > approach, highlighting its **broader implications beyond the technical
> > > > > > aspects**:
> > > > > >
> > > > > > 1.  Although many studies have explored contamination detection of LLMs,
> > > > > >     we further deepen the application of these methods by integrating a
> > > > > >     blocking strategy to restore the performance of contaminated LLMs.
> > > > > >
> > > > > > 2.  While there have been numerous works on contamination-free
> > > > > >     performance evaluation methods for models, some[4] require the creation
> > > > > >     of new uncontaminated datasets, which is highly resource-intensive,
> > > > > >     while others[5] rely on using other LLMs for dynamic evaluation, making
> > > > > >     it difficult to establish a static and consistent leaderboard. Only
> > > > > >     a few studies have conducted contamination-free performance
> > > > > >     evaluations using existing datasets, and these methods often involve
> > > > > >     extensive preprocessing[6] or multiple sampling iterations[7], making them
> > > > > >     highly time-consuming.
> > > > > >
> > > > > >     LNE-Blocking is the first strategy that evaluates the performance of
> > > > > >     contaminated models through two inferences without modifying the
> > > > > >     original dataset.
> > > > > >
> > > > > > As such, we are
> > > > > > wondering whether there is anything else we could do or explain to
> > > > > > increase your appraisal beyond "rejection". If there are other things we
> > > > > > could do during the discussion period or before publication that you
> > > > > > think would substantially improve the manuscript, please let us know.
> > > > > >
> > > > > > >[1] Raymond Li, Loubna Ben Allal, Yangtian Zi, Niklas Muennighoff, Denis Kocetkov, Chenghao Mou, Marc Marone, Christopher Akiki, Jia Li, Jenny Chim, et al. Starcoder: may the source be with you! arXiv preprint arXiv:2305.06161, 2023b.
> > > > > > >
> > > > > > >[2] Qintong Li, Leyang Cui, Xueliang Zhao, Lingpeng Kong, and Wei Bi. Gsm-plus: A comprehensive benchmark for evaluating the robustness of llms as mathematical problem solvers. arXiv preprint arXiv:2402.19255, 2024b.
> > > > > > >
> > > > > > > [3] Xuezhi Wang and Denny Zhou. Chain-of-thought reasoning without prompting. arXiv preprint
> > > > > > arXiv:2402.10200, 2024.
> > > > > > >
> > > > > > >[4] Hugh Zhang, Jeff Da, Dean Lee, Vaughn Robinson, Catherine Wu, Will Song, Tiffany Zhao, Pranav Raja, Dylan Slack, Qin Lyu, et al. A careful examination of large language model performance on grade school arithmetic. arXiv preprint arXiv:2405.00332, 2024.
> > > > > > >
> > > > > > >[5] Xiang Li, Yunshi Lan, and Chao Yang. Treeeval: Benchmark-free evaluation of large language models through tree planning. arXiv preprint arXiv:2402.13125, 2024c.
> > > > > > >
> > > > > > >[6] Wenhong Zhu, Hongkun Hao, Zhiwei He, Yunze Song, Yumeng Zhang, Hanxu Hu, Yiran Wei, Rui Wang, and Hongyuan Lu. Clean-eval: Clean evaluation on contaminated large language models. arXiv preprint arXiv:2311.09154, 2023.
> > > > > > >
> > > > > > >[7] Yihong Dong, Xue Jiang, Huanyu Liu, Zhi Jin, Bin Gu, Mengfei Yang, and Ge Li. Generalization or memorization: Data contamination and trustworthy evaluation for large language models. In Lun-Wei Ku, Andre Martins, and Vivek Srikumar (eds.), Findings of the Association for Computational Linguistics ACL 2024, pp. 12039–12050, Bangkok, Thailand and virtual meeting, August 2024. Association for Computational Linguistics. doi: 10.18653/v1/2024.findings-acl.716. URL https://aclanthology.org/2024.findings-acl.716.

---

### Official Review · Reviewer_usW6 · 2024-11-03

**Soundness:** 2
**Presentation:** 3
**Contribution:** 2
**Rating:** 5
**Confidence:** 2

**Summary:**

This paper tackles the critical issue of data contamination in LLMs, where training data unintentionally includes evaluation benchmarks, complicating fair model assessment. The authors propose LNE-Blocking, a two-part framework that first detects contaminated data using Length Normalized Entropy (LNE) and then mitigates contamination effects by dynamically adjusting blocking during text generation. This approach not only provides state-of-the-art detection but also enables efficient and fair evaluation of LLMs, requiring only two inferences for contamination mitigation. Robust across contamination levels, LNE-Blocking offers a practical and effective solution to this growing challenge in LLM development.

**Strengths:**

- The paper addresses a highly significant topic.
- The writing is well-structured and clearly organized.
- The proposed method demonstrates originality.

**Weaknesses:**

- **Concerns Regarding the Blocking Method**: While I appreciate the idea that "similar to how humans can rephrase their thoughts when interrupted, LLMs have the potential to generate alternative answers when their default response is blocked," there are practical concerns with the chosen approach for blocking tokens. Specifically, it may lead to issues if unique tokens at certain positions are blocked, making it difficult to generate correct answers. For instance, a long variable name might be split into several consecutive tokens, and blocking a critical unique token in the middle could result in inaccurate code generation.

- **Lack of Descriptions for Key Information**: Certain key terms and settings lack clear definitions, such as "Fixed Blocking 1, 2, and 5," which appear in Figure 1 and throughout the experiments. Further clarification is needed on what these terms specifically refer to.

- **Minor Typos**: There are small typographical issues, such as mismatched quotation marks in Figure 1's caption ("Cont."), which should be addressed.

- **Limited Experimental Evaluation**: It appears that the effectiveness of the DATA CONTAMINATION DETECTION approach was only evaluated on a single model. Additionally, the results suggest that the proposed LNE method outperformed the baseline only under Mild Cont. conditions, with limited improvement. To demonstrate the broader effectiveness of the approach, evaluations on additional models are necessary.

**Questions:**

1. How does the proposed blocking algorithm ensure that critical, unique tokens essential to answering the question are not blocked?
2. What are the specific definitions of Fixed Blocking 1, 2, and 5?
3. Could the authors provide additional evaluation results on DATA CONTAMINATION DETECTION across a broader range of models?

---

> ### Author Response · Authors · 2024-12-01
> **Weakness one, two and three**
>
> Thank you sincerely for your thorough reading of our work and detailed
> suggestions for improvements. We are glad that the paper addresses a
> highly significant topic and that the proposed method demonstrates
> originality.
>
> We would like to address your concerns in the following.
>
> ## Weakness one:
>
> Thank you for your valuable feedback. Considering this factor, we
> perform the blocking operation starting from the generation of the first
> token and continue until the required number of blocking operations for
> the current sample is reached, as described in section 4.3.1 of the revised paper. This
> ensures that subsequent tokens use the newly generated variable names,
> thereby maintaining consistency throughout the output.
>
> Additionally, we have included a heatmap of token replacement frequency
> during the blocking operation in the Appendix C of the revised paper.
> Below are some examples for your reference.
>
> Replace _if with _return:
> ```python
> if not numbers:
>     return 0.0
> mean = sum(numbers) / len(numbers)
> return sum(abs(x - mean) for x in numbers) / len(numbers)
>
> ↓↓↓
>
> return float((sum(abs(x - mean(numbers)) for x in numbers)) / len(numbers))
> ```
> Replace _return with _ret:
> ```python
> return [x + 1 for x in l]
>
> ↓↓↓
>
> ret_l=[]
> for i in l:
>     ret_l.append(i+1)
> return ret_l
> ```
> Replace _return with return:
> ```python
> return number % 1.0
>
> ↓↓↓
>
> returnnumber=number%1
> return return number
> ```
>
> ## Weakness two:
>
> Thank you for pointing this out. We apologize for the lack of clarity.
> The terms "Fixed Blocking 1, 2, and 5" refer to the fixed number of
> blocking operations applied for any given question, as opposed to the
> dynamic approach where the count of blocking operations is determined
> based on contamination detection scores for specific samples. Now, we
> provide a clear definition of these terms in the revised version of the
> paper.
>
> ## Weakness three:
> Thank you for pointing this out. We have reviewed and
> addressed some typographical issues, including the mismatched quotation
> marks in the caption of Figure 1 ("Cont.").

---

> > ### Author Response · Authors · 2024-12-01
> > **Weakness Four**
> >
> > ## Weakness Four:
> > 1.  "LNE Only perform well in Mild Cont."
> >
> >     **a.** As shown in Table 1 of the paper, all methods perform
> >     significantly worse on the mildly contaminated model compared to the
> >     moderately and heavily contaminated models, indicating that
> >     pollution detection is more challenging in the mild contamination
> >     scenario. In this challenging task, our proposed LNE strategy
> >     outperforms other methods by 5 percentage points in both F1 score
> >     and AUC. However, in the moderate and heavy contamination scenarios,
> >     where the task is simpler, our method performs only 1 percentage
> >     point worse than Min-k. This results in overall superior performance
> >     of our strategy.
> >
> >     **b.** Since the contamination detection strategy needs to be
> >     integrated with our subsequent contamination mitigation evaluation
> >     strategy, and the contamination level of the current model is
> >     unknown when applying the contamination mitigation evaluation
> >     strategy, it requires a contamination detection method that performs
> >     well overall to ensure better performance in the task of
> >     contamination mitigation evaluation. We have added this experiment,
> >     as shown in the table below, and the results demonstrate that using
> >     LNE leads to improved performance in the contamination mitigation
> >     evaluation task compared to Min-k.
> >
> >     | Strategy      | Mild Cont.    | Moderate Cont. | Heavy Cont.   | Average        |
> >     |---------------|---------------|----------------|---------------|----------------|
> >     | PPL-Blocking  | 0.3 (0.044)   | 0.28 (0.037)   | 0.274 (0.047) | 0.284 (0.042)  |
> >     | MINK-Blocking | 0.323 (0.058) | 0.274 (0.051)  | 0.27 (**0.043**)  | 0.29 (0.052)   |
> >     | LNE-Blocking  | 0.297 (**0.035**) | 0.282 (**0.032**)  | 0.271 (0.045) | 0.283 (**0.037**)  |
> >
> >     **c.** Our motivation is to treat contamination detection as a
> >     foundational task, with the primary focus on solving the
> >     contamination mitigation evaluation task. As a result, we just
> >     selected a more effective contamination detection strategy without
> >     conducting further performance exploration.
> >
> > 2.  According to your advice: "Evaluations on additional models are necessary." We
> >     have supplemented the performance experiments of these strategies on
> >     three additional models, as shown in the table below. Our approach still
> >     outperforms the single-inference-based methods, and only slightly
> >     underperforms CDD in certain cases. It is worth noting that CDD
> >     requires 50 sampling iterations for contamination detection, while
> >     our method operates with just a single inference. We have also
> >     included this additional information in the appendix of the revised
> >     paper.
> >
> >     | Model     | Strategy   | Mild Cont. | Mild Cont. | Moderate Cont. | Moderate Cont. | Heavy Cont. | Heavy Cont. | Overall  | Overall  |
> >     |-----------|------------|------------|------------|----------------|----------------|-------------|-------------|----------|----------|
> >     |           |            | F1 score   | AUC        | F1 score       | AUC            | F1 score    | AUC         | F1 score | AUC      |
> >     | Llama 2   | Min-k      | 0.701      | 0.579      | 0.859          | 0.890          | 0.921       | 0.967       | 0.774    | 0.820    |
> >     | Llama 2   | Perplexity | 0.727      | 0.635      | 0.875          | 0.928          | 0.940       | 0.984       | 0.788    | 0.857    |
> >     | Llama 2   | CDD        | 0.712      | **0.742**      | 0.824          | 0.887          | 0.935       | 0.972       | 0.798    | **0.869**    |
> >     | Llama 2   | LNE        | **0.738**      | 0.648      | **0.884**          | **0.932**          | **0.942**       | **0.986**       | **0.800**    | 0.863    |
> >     | Llama 3.1 | Min-k      | 0.728      | 0.681      | 0.930          | 0.967          | 0.962       | 0.992       | 0.830    | 0.889    |
> >     | Llama 3.1 | Perplexity | **0.745**      | **0.709**      | 0.944          | 0.983          | 0.972       | 0.997       | 0.839    | **0.905**    |
> >     | Llama 3.1 | CDD        | 0.667      | 0.608      | 0.891          | 0.936          | 0.959       | 0.988       | 0.805    | 0.853    |
> >     | Llama 3.1 | LNE        | 0.744      | 0.707      | **0.952**          | **0.985**          | **0.979**       | **0.998**       | **0.844**    | **0.905**    |
> >     | CodeGen   | Min-k      | 0.736      | 0.700      | 0.833          | 0.892          | 0.878       | 0.934       | 0.778    | 0.847    |
> >     | CodeGen   | Perplexity | 0.732      | 0.710      | 0.833          | 0.895          | 0.907       | 0.949       | 0.771    | 0.856    |
> >     | CodeGen   | CDD        | 0.686      | 0.707      | **0.867**          | **0.915**          | **0.963**       | **0.992**       | **0.813**    | **0.876**    |
> >     | CodeGen   | LNE        | **0.741**      | **0.725**      | 0.827          | 0.905          | 0.914       | 0.958       | 0.777    | 0.867    |

---

> > > ### Author Response · Authors · 2024-12-01
> > > **Novelty Beyond Technique**
> > >
> > > ## Novelty Beyond Technique
> > > **Last but importantly**, we would like to emphasize the novelty of our
> > > approach, highlighting its **broader implications beyond the technical
> > > aspects**:
> > >
> > > 1.  Although many studies have explored contamination detection of LLMs,
> > >     we further deepen the application of these methods by integrating a
> > >     blocking strategy to restore the performance of contaminated LLMs.
> > >
> > > 2.  While there have been numerous works on contamination-free
> > >     performance evaluation methods for models, some[1] require the creation
> > >     of new uncontaminated datasets, which is highly resource-intensive,
> > >     while others[2] rely on using other LLMs for dynamic evaluation, making
> > >     it difficult to establish a static and consistent leaderboard. Only
> > >     a few studies have conducted contamination-free performance
> > >     evaluations using existing datasets, and these methods often involve
> > >     extensive preprocessing[3] or multiple sampling iterations[4], making them
> > >     highly time-consuming.
> > >
> > >     LNE-Blocking is the first strategy that evaluates the performance of
> > >     contaminated models through two inferences without modifying the
> > >     original dataset.
> > >
> > > As such, we are
> > > wondering whether there is anything else we could do or explain to
> > > increase your appraisal beyond "weak rejection". If there are other
> > > things we could do during the discussion period or before publication
> > > that you think would substantially improve the manuscript, please let us
> > > know.
> > >
> > > ### References
> > > >[1] Hugh Zhang, Jeff Da, Dean Lee, Vaughn Robinson, Catherine Wu, Will Song, Tiffany Zhao, Pranav Raja, Dylan Slack, Qin Lyu, et al. A careful examination of large language model performance on grade school arithmetic. arXiv preprint arXiv:2405.00332, 2024.
> > > >
> > > >[2] Xiang Li, Yunshi Lan, and Chao Yang. Treeeval: Benchmark-free evaluation of large language models through tree planning. arXiv preprint arXiv:2402.13125, 2024c.
> > > >
> > > >[3] Wenhong Zhu, Hongkun Hao, Zhiwei He, Yunze Song, Yumeng Zhang, Hanxu Hu, Yiran Wei, Rui Wang, and Hongyuan Lu. Clean-eval: Clean evaluation on contaminated large language models. arXiv preprint arXiv:2311.09154, 2023.
> > > >
> > > >[4] Yihong Dong, Xue Jiang, Huanyu Liu, Zhi Jin, Bin Gu, Mengfei Yang, and Ge Li. Generalization or memorization: Data contamination and trustworthy evaluation for large language models. In Lun-Wei Ku, Andre Martins, and Vivek Srikumar (eds.), Findings of the Association for Computational Linguistics ACL 2024, pp. 12039–12050, Bangkok, Thailand and virtual meeting, August 2024. Association for Computational Linguistics. doi: 10.18653/v1/2024.findings-acl.716. URL https://aclanthology.org/2024.findings-acl.716.

---

### Official Review · Reviewer_Vu4w · 2024-11-05

**Soundness:** 3
**Presentation:** 1
**Contribution:** 2
**Rating:** 3
**Confidence:** 2

**Summary:**

The article deals with the problem of training set contamination in which one would like to validate that an LLM being evaluated was not trained with a given test sample. The proposed contamination detection method relies on the entropy of output tokens normalized by the output length. Following the detection mechanism the authors propose a method to evaluate strategies for mitigating the contamination techniques. The contamination detection is benchmarked against three competitors proposed in 2023-2024. The contamination mitigation method is benchmarked against one competitor published in 2024. The results show that the performance of the proposed methods is favorable under certain conditions.

**Strengths:**

*  Benchmarking against very recent models


* Good motivation

**Weaknesses:**

* Marginal performance benefit

Results reported in Tale 1 do not provide a sufficient margin over Min-k. Overall LNE does not significantly outperform Min-k.

In table 2, what about the other bold numbers? Is TED significantly outperforms LNE-Blocking  on CodeGen Mild cont?


* Missing theoretical justification of the contamination detection approach

It is not clear why is the input x necessary when defining LNE if it is not used in Eq 1.
y is defined as the output and l is defined as its length. Yet the last token in the sequence y_1,...y_N is indexed by N. Both N and l are used in Eq 1 making its interpretation challenging.

According to eq 2 LNE signals when the token probabilities are skewed. Why should LNE work better than Min-k% Prob for detecting contamination?


* Readability and organization can be improved

It is not clear what "extensive SOTA results" means in the second contribution bullet. Earlier, the authors refer to SOTA performance, implying that the proposed approach is now the best. In this context, the term "extensive" is not clear.

... whether this data has been trained by the model M. --> whether this data was used to train model M. OR .. whether model M was trained using this data.

Data and subject models are described three times in three different places: Sections 4.1 4.2, Section 5.1 first par. Section 5.2.1 first par

The explanations of the baselines on page 6 lack accuracy and important details. For example, it is stated that Min-k% Prob computes the probability of k% least probable tokens, but (1) it is not written that the low total prob indicates contamination, (2) the relationship between the reference answer and the prompt mentioned in that statement is not clear. The same goes for perplexity and CDD.

The related work section is expected to elaborate on the main competitors. Currently, they are missing from section 6.  When they are added the section should appear earlier in the paper before the baselines are actually used.



*  Benchmark is based on an unvalidated assumption that LLMs were not contaminated by the HumanEval dataset (line 269).

* Missing some experimental details

It is not clear how the thresholds were chosen for all methods for computing F1 in Table 1. Unfairly chosen thresholds can bias the results. Luckily this problem does not affect AUC.

**Questions:**

Please provide a theoretical justification explaining why LNE should work better than Min-k% Prob for detecting contamination.


Why is contamination detection tested only on CodeLlama and not all four models?

---

> ### Author Response · Authors · 2024-12-01
> **Weakness one**
>
> Thank you sincerely for your thorough reading of our work and detailed
> suggestions for improvements. We are glad that you agree that we have a
> good motivation and benchmark against very recent models.
>
> We would like to address your concerns in the following.
>
> ### Weakness one:
>
> 1.  **a.** As shown in Table 1 of the paper, all methods perform
>     significantly worse on the mildly contaminated model compared to the
>     moderately and heavily contaminated models, indicating that
>     pollution detection is more challenging in the mild contamination
>     scenario. In this challenging task, our proposed LNE strategy
>     outperforms other methods by 5 percentage points in both F1 score
>     and AUC. However, in the moderate and heavy contamination scenarios,
>     where the task is simpler, our method performs only 1 percentage
>     point worse than Min-k. This results in overall superior performance
>     of our strategy.
>
>     **b.** Since the contamination detection strategy needs to be
>     integrated with our subsequent contamination mitigation evaluation
>     strategy, and the contamination level of the current model is
>     unknown when applying the contamination mitigation evaluation
>     strategy, it requires a contamination detection method that performs
>     well overall to ensure better performance in the task of
>     contamination mitigation evaluation. We have added this experiment,
>     as shown in the table below, and the results demonstrate that using
>     LNE leads to improved performance in the contamination mitigation
>     evaluation task compared to Min-k.
>
>     | Strategy      | Mild Cont.    | Moderate Cont. | Heavy Cont.   | Average        |
>     |---------------|---------------|----------------|---------------|----------------|
>     | PPL-Blocking  | 0.3 (0.044)   | 0.28 (0.037)   | 0.274 (0.047) | 0.284 (0.042)  |
>     | MINK-Blocking | 0.323 (0.058) | 0.274 (0.051)  | 0.27 (**0.043**)  | 0.29 (0.052)   |
>     | LNE-Blocking  | 0.297 (**0.035**) | 0.282 (**0.032**)  | 0.271 (0.045) | 0.283 (**0.037**)  |
>
>     **c.** Our motivation is to treat contamination detection as a
>     foundational task, with the primary focus on solving the
>     contamination mitigation evaluation task. As a result, we just
>     selected a more effective contamination detection strategy without
>     conducting further performance exploration.
>
> 2.  **a.** We apologize for any confusion caused. The bold numbers in
>     Table 2 indicate the best-performing contamination mitigation
>     strategy for each contamination level. A smaller value indicates
>     better contamination mitigation evaluation strategy, closer to the
>     performance of the uncontaminated model after applying the
>     contamination mitigation strategy to contaminated model.
>
>     **b.** As shown in Table 2, TED on CodeGen with mild contamination
>     outperforms LNE-Blocking by only 4 percentage points. However, for
>     moderate and heavy contamination, LNE-Blocking demonstrates superior
>     performance, leading to better overall performance. Additionally, we
>     have underlined significantly better strategies. For example,
>     LNE-Blocking outperforms TED by nearly 10 percentage points on
>     CodeLlama with moderate and heavy contamination, as well as on
>     Llama3.1 with heavy contamination.

---

> > ### Author Response · Authors · 2024-12-01
> > **Weakness two, three and four**
> >
> > ### Weakness two:
> >
> > 1.  Thank you for your valuable feedback.
> >
> >     **a.** As you pointed out, the input *x* is unnecessary when
> >     defining LNE, as it is not used in Equation 1. Our initial intention
> >     was to express that *y* is the output of the LLM corresponding to
> >     *x*, not the reference answer. We have clarified this point in the
> >     revised version of the paper.
> >
> >     **b.** *l* and *N* were used interchangeably, which caused
> >     confusion. We apologize for this oversight and have corrected it in
> >     the updated version.
> >
> > 2.  I believe this is due to LNE utilizing more comprehensive
> >     information compared to Min-k. While Min-k primarily focuses on the
> >     probabilities of the selected tokens at some position in the output
> >     sequence, LNE incorporates not only the probabilities of the
> >     selected tokens but also the probabilities of unselected tokens.
> >     This enables LNE to capture a broader view of the entire probability
> >     distribution, providing a more complete representation of the
> >     model’s behavior. As a result, it may be more effective in detecting
> >     contamination.
> >
> > ### Weakness three:
> >
> > 1.  "Readability and organization can be improved": Thank you for your
> >     thoughtful and detailed feedback. We have made the following
> >     revisions in the revised manuscript in response to your comments:
> >
> >     **a.** "Extensive SOTA results": We have clarified the term
> >     "extensive" in the revised version to explicitly describe the wide
> >     range of experiments and datasets used to demonstrate the
> >     superiority of our approach over existing SOTA methods.
> >
> >     **b.** Phrasing of "whether this data has been trained by model M":
> >     We have updated this phrasing to the clearer expression, "whether
> >     this data was used to train the model M".
> >
> >     **c.** Repetition of data and subject models: We have streamlined
> >     the descriptions of data and subject models to avoid redundancy
> >     across Sections 5.1, 5.2, 6.1, and 6.2.1 in the revised paper.
> >
> >     **d.** We have revised the baseline explanations on page 6 to
> >     include the missing details, such as the link between the value of
> >     Min-k% Prob and contamination, and clarified the relationship
> >     between the reference answer and the prompt. For perplexity and CDD,
> >     we have provided more precise explanations.
> >
> >     **e.** The Related Work section has been expanded to include works
> >     from related directions, with an emphasis on highlighting the
> >     distinctions between our approach and previous methods. And we have
> >     moved it earlier in the paper to provide a clearer context before
> >     introducing the baselines. Furthermore, Appendix D elaborates on the
> >     main competitors.
> > ### Weakness Four:
> >
> > 1.  Thank you for your insightful comment. Indeed, the assumption that
> >     LLMs were not contaminated by the HumanEval dataset is based
> >     primarily on the statements made in the corresponding model papers.
> >     Given the large computational cost of training these models, we made
> >     an approximation by relying on this assumption.
> >
> >     To address this concern, we have introduced a new dataset, GSM-Plus [1],
> >     which is an augmented version of GSM8K with various mathematical
> >     perturbations, including numerical variation, arithmetic variation,
> >     problem understanding challenges, distractor insertion, and critical
> >     thinking tasks. This dataset was released in January 2024, which is
> >     after the release of Llama2. Given the randomness of these
> >     perturbations and the innovative nature of the techniques employed,
> >     it is highly likely that the original uncontaminated version of
> >     Llama2 was not exposed to contamination during its training.
> >
> >     We have further validated the effectiveness of the LNE-Blocking
> >     strategy on this new dataset, as shown in the table below.
> >
> >     | Model     | Strategy     | Mild Cont.    | Moderate Cont. | Heavy Cont.   | Average        |
> >     |-----------|--------------|---------------|----------------|---------------|----------------|
> >     | Llama 2   | Sampling     | 0.243         | 0.683          | 0.752         | 0.542          |
> >     | Llama 2   | Greedy       | 0.178         | 0.651          | 0.747         | 0.505          |
> >     | Llama 2   | TED          | 0.235 (0.084) | 0.305 (0.154)  | 0.143 (**0.008**) | 0.235 (0.089)  |
> >     | Llama 2   | LNE-Blocking | 0.112 (**0.023**) | 0.14 (**0.051**)   | 0.139 (0.049) | 0.13 (**0.04**)    |
> >     | Llama 3.1 | Sampling     | 0.663         | 0.947          | 0.982         | 0.853          |
> >     | Llama 3.1 | Greedy       | 0.513         | 0.917          | 0.982         | 0.788          |
> >     | Llama 3.1 | TED          | 0.655 (**0.068**) | 0.378 (0.275)  | 0.119 (0.467) | 0.408 (0.252)  |
> >     | Llama 3.1 | LNE-Blocking | 0.341 (0.085) | 0.315 (**0.111**)  | 0.327 (**0.099**) | 0.328 (**0.098**)  |
> >
> >     We have also included
> >     this additional information in the Appendix B of the revised paper.

---

> > > ### Author Response · Authors · 2024-12-01
> > > **Weakness Five, Question one and Question two**
> > >
> > > ### Weakness Five:
> > >
> > > 1.  In the contamination detection task, when calculating the F1 score,
> > >     we select the optimal threshold for each strategy at different
> > >     contamination levels to obtain the best F1 for each strategy. This
> > >     ensures a fair comparison of the F1 scores across the strategies. We
> > >     have also addressed this point in the revised version of the paper.
> > >     Thank you for pointing out this issue.
> > >
> > > ### Question one:
> > >
> > > Min-k% Prob and LNE are illustrated as:
> > > $$
> > > \text{Min-k} (y) = -\frac{1}{E} \sum_{y_i \in \text{Min-k Set}(y) }\log p(y_i)
> > > $$
> > > $$
> > > \mathrm{LNE (y)} = -\frac{1}{N} \sum_{i=1}^N\sum_j^{V} p\left (y_i=j\right) \log p\left (y_i=j\right).
> > > $$
> > > where *y* is the greedy decoded output, Min-k Set(y) is
> > > the set of k% least probable tokens in the output, *E* is the size of
> > > the Min-k Set(y) set, and *N* is the length of the output.
> > >
> > > As these equations show, LNE utilizes more information compared to
> > > Min-k. While Min-k focuses only on the probabilities of the selected
> > > tokens at some position in the output sequence, LNE considers not only
> > > the probabilities of the selected tokens but also the probabilities of
> > > the unselected tokens at each position in the output. This allows LNE to
> > > capture a more comprehensive representation of the probability
> > > distribution, making it more effective at detecting contamination.
> > >
> > > ### Question two:
> > >
> > > **a.** Our motivation is to treat contamination detection as a
> > > foundational task, with the primary focus on solving the contamination
> > > mitigation evaluation task. As a result, we just selected a more
> > > effective contamination detection strategy without conducting extensive
> > > experiments for the task.
> > >
> > > **b.** We have supplemented the performance experiments of these
> > > strategies on three additional models, as shown in the table below. Our approach still
> > > outperforms the single-inference-based methods, and only slightly
> > > underperforms CDD in certain cases. It is worth noting that CDD requires
> > > 50 sampling iterations for contamination detection, while our method
> > > operates with just a single inference. We have also included this
> > > additional information in the Appendix A of the revised paper.
> > >
> > > | Model     | Strategy   | Mild Cont. | Mild Cont. | Moderate Cont. | Moderate Cont. | Heavy Cont. | Heavy Cont. | Overall  | Overall  |
> > > |-----------|------------|------------|------------|----------------|----------------|-------------|-------------|----------|----------|
> > > |           |            | F1 score   | AUC        | F1 score       | AUC            | F1 score    | AUC         | F1 score | AUC      |
> > > | Llama 2   | Min-k      | 0.701      | 0.579      | 0.859          | 0.890          | 0.921       | 0.967       | 0.774    | 0.820    |
> > > | Llama 2   | Perplexity | 0.727      | 0.635      | 0.875          | 0.928          | 0.940       | 0.984       | 0.788    | 0.857    |
> > > | Llama 2   | CDD        | 0.712      | **0.742**      | 0.824          | 0.887          | 0.935       | 0.972       | 0.798    | **0.869**    |
> > > | Llama 2   | LNE        | **0.738**      | 0.648      | **0.884**          | **0.932**          | **0.942**       | **0.986**       | **0.800**    | 0.863    |
> > > | Llama 3.1 | Min-k      | 0.728      | 0.681      | 0.930          | 0.967          | 0.962       | 0.992       | 0.830    | 0.889    |
> > > | Llama 3.1 | Perplexity | **0.745**      | **0.709**      | 0.944          | 0.983          | 0.972       | 0.997       | 0.839    | **0.905**    |
> > > | Llama 3.1 | CDD        | 0.667      | 0.608      | 0.891          | 0.936          | 0.959       | 0.988       | 0.805    | 0.853    |
> > > | Llama 3.1 | LNE        | 0.744      | 0.707      | **0.952**          | **0.985**          | **0.979**       | **0.998**       | **0.844**    | **0.905**    |
> > > | CodeGen   | Min-k      | 0.736      | 0.700      | 0.833          | 0.892          | 0.878       | 0.934       | 0.778    | 0.847    |
> > > | CodeGen   | Perplexity | 0.732      | 0.710      | 0.833          | 0.895          | 0.907       | 0.949       | 0.771    | 0.856    |
> > > | CodeGen   | CDD        | 0.686      | 0.707      | **0.867**          | **0.915**          | **0.963**       | **0.992**       | **0.813**    | **0.876**    |
> > > | CodeGen   | LNE        | **0.741**      | **0.725**      | 0.827          | 0.905          | 0.914       | 0.958       | 0.777    | 0.867    |

---

> ### Author Response · Authors · 2024-12-01
> **Novelty Beyond Technique**
>
> ## Novelty Beyond Technique
> **Last but importantly**, we would like to emphasize the novelty of our
> approach, highlighting its **broader implications beyond the technical
> aspects**:
>
> 1.  Although many studies have explored contamination detection of LLMs,
>     we further deepen the application of these methods by integrating a
>     blocking strategy to restore the performance of contaminated LLMs.
>
> 2.  While there have been numerous works on contamination-free
>     performance evaluation methods for models, some[2] require the creation
>     of new uncontaminated datasets, which is highly resource-intensive,
>     while others[3] rely on using other LLMs for dynamic evaluation, making
>     it difficult to establish a static and consistent leaderboard. Only
>     a few studies have conducted contamination-free performance
>     evaluations using existing datasets, and these methods often involve
>     extensive preprocessing[4] or multiple sampling iterations[5], making them
>     highly time-consuming.
>
>     LNE-Blocking is the first strategy that evaluates the performance of
>     contaminated models through two inferences without modifying the
>     original dataset.
>
> As such, we are wondering whether there is anything else we could do or
> explain to increase your appraisal beyond "rejection". If there are
> other things we could do during the discussion period or before
> publication that you think would substantially improve the paper, please
> let us know.
>
> ### References
> >[1] Qintong Li, Leyang Cui, Xueliang Zhao, Lingpeng Kong, and Wei Bi. Gsm-plus: A comprehensive benchmark for evaluating the robustness of llms as mathematical problem solvers. arXiv preprint arXiv:2402.19255, 2024b.
> >
> >[2] Hugh Zhang, Jeff Da, Dean Lee, Vaughn Robinson, Catherine Wu, Will Song, Tiffany Zhao, Pranav Raja, Dylan Slack, Qin Lyu, et al. A careful examination of large language model performance on grade school arithmetic. arXiv preprint arXiv:2405.00332, 2024.
> >
> >[3] Xiang Li, Yunshi Lan, and Chao Yang. Treeeval: Benchmark-free evaluation of large language models through tree planning. arXiv preprint arXiv:2402.13125, 2024c.
> >
> >[4] Wenhong Zhu, Hongkun Hao, Zhiwei He, Yunze Song, Yumeng Zhang, Hanxu Hu, Yiran Wei, Rui Wang, and Hongyuan Lu. Clean-eval: Clean evaluation on contaminated large language models. arXiv preprint arXiv:2311.09154, 2023.
> >
> >[5] Yihong Dong, Xue Jiang, Huanyu Liu, Zhi Jin, Bin Gu, Mengfei Yang, and Ge Li. Generalization or memorization: Data contamination and trustworthy evaluation for large language models. In Lun-Wei Ku, Andre Martins, and Vivek Srikumar (eds.), Findings of the Association for Computational Linguistics ACL 2024, pp. 12039–12050, Bangkok, Thailand and virtual meeting, August 2024. Association for Computational Linguistics. doi: 10.18653/v1/2024.findings-acl.716. URL https://aclanthology.org/2024.findings-acl.716.

---

### Meta-Review · Area_Chair_2eWq · 2024-12-19

**Metareview:**

Data contamination is a phenomenon where data from a benchmark appears in the training data of an LLM, resulting in an unfair advantage of the LLM when being evaluated. The paper provides a method to both identify and mitigate data contamination. The reviews agree that this problem is highly motivated, and appreciated the discussion related to its motivation given in the paper. The approach was mentioned to be novel and the baselines are from recent and state of the art results.

The main concerns raised are: (1) Narrow experimental analysis: D5E5 mentioned the need for more experiments, as well as more realistic test cases, usW6 required an evaluation on more models. (2) writing clarity: All reviewers mentioned issues with the technical writing: vaguely defined notions, unclear information, missing experiment details, etc. The authors replied with some new experiments and clarifications for the unclear issues mentioned by the reviews. This reply mitigates some of the concerns but not all. Moreover, the work required to integrate the new experiments and improve the writing quality seems too extensive to be done without having another round of reviews.

**Additional Comments On Reviewer Discussion:**

see meta review

---

### Decision · Program_Chairs · 2025-01-22

Reject